# Physical and chemical properties of black carbon and organic matter from different combustion and photochemical sources using aerodynamic aerosol classification

Dawei Hu[1,*], M. Rami Alfarra[1,2,#], Kate Szpek[3], Justin M. Langridge[3], Michael I. Cotterell[5], Claire Belcher[4], Ian Rule[4], Zixia Liu[4], Chenjie Yu[1], Yunqi Shao[1], Aristeidis Voliotis[1], Mao Du[1], Brett Smith[6], Greg Smallwood[6], Prem Lobo[6], Dantong Liu[7], Jim M. Haywood[4], Hugh Coe[1], James D. Allan[1,2,*]

[1]Department of Earth and Environmental Sciences, University of Manchester, UK.

[2]National Centre for Atmospheric Science, University of Manchester, Manchester, UK.

[3]Observation Based Research, Met Office, Exeter, UK

[4]College for Engineering, Mathematics and Physical Sciences, University of Exeter, Exeter, UK

[5]School of Chemistry, University of Bristol, Bristol, UK

[6]Metrology Research Centre, National Research Council Canada, Ottawa, Canada.

[7]Department of Atmospheric Sciences, School of Earth Sciences, Zhejiang University, Hangzhou, Zhejiang, China.

[#]Currently at Qatar Environment and Energy Research Institute (QEERI), Hamad Bin Khalifa University (HBKU), Doha, Qatar.

[*]Correspondence to: dawei.hu@manchester.ac.uk; james.allan@manchester.ac.uk

**Abstract.**

The physical and chemical properties of black carbon (BC) and organic aerosols are important for predicting their radiative forcing in the atmosphere. During the Soot Aerodynamic Size Selection for Optical properties (SASSO) project and a EUROCHAMP-2020 transnational access project, different types of light absorbing carbon were studied, including BC from catalytically stripped diesel exhaust, an inverted flame burner, a colloidal graphite standard (Aquadag), and from controlled flaming wood combustion. Brown carbon (BrC) was also investigated in the form of organic aerosol emissions from wood burning

(pyrolysis and smouldering) and from the nitration of secondary organic aerosol (SOA) proxies produced in a photochemical reaction chamber. Here we present insights into the physical and chemical properties of the aerosols, with optical properties presented in subsequent publications. The dynamic shape factor ($\chi$) of BC particles and material density ($\rho_m$) of organic aerosols were investigated by coupling a charging-free Aerodynamic Aerosol Classifier (AAC) with a Centrifugal Particle Mass Analyzer (CPMA) and Scanning Mobility Particle Sizer (SMPS). The morphology of BC particles was captured by transmission electron microscopy (TEM). For BC particles from the diesel engine and flame burner emissions, the primary spherule sizes were similar, around 20 nm. With increasing particle size, BC particles adopted more collapsed/compacted morphologies for the former source but tended to show more aggregated morphologies for the latter source. For particles emitted from the combustion of dry wood samples, the $\chi$ of BC particles and the $\rho_m$ of organic aerosols were observed in the ranges 1.8-2.17 and 1.22-1.32 g/cm$^3$, respectively. Similarly, for wet wood samples, the $\chi$ and $\rho_m$ ranges were 1.2-1.85 and 1.44-1.60 g/cm$^3$, respectively. Aerosol mass spectrometry measurements show no clear difference in mass spectra of the organic aerosols in individual burn phases (pyrolysis or smouldering phase) with the moisture content of the wood samples. This suggests that the effect moisture has on the organic chemical profile of wood burning emissions is through changing the durations of the different phases of the burn cycle, not through the chemical modification of the individual phases. In this study, the incandescence signal of a Single Particle Soot Photometer (SP2) was calibrated with three different types of BC particles and compared with that from an Aquadag standard that is commonly used to calibrate SP2 incandescence to a BC mass. A correction factor is defined as the ratio of the incandescence signal from an alternative BC source to that from the Aquadag standard, and took values of 0.821 ± 0.002 (or 0.794 ± 0.005), 0.879 ± 0.003 and

0.843 ± 0.028 to 0.913 ± 0.009 for the BC particles emitted from the diesel engine running under hot (or cold idle) conditions, the flame burner and wood combustion, respectively. These correction factors account for differences in instrument response to BC from different sources compared to the standardised Aquadag calibration and are more appropriate than the common value of 0.75 recommended by Laborde et al. (2012b) when deriving the mass concentration of BC emitted from diesel engines. Quantifying the correction factor for many types of BC particles found commonly in the atmosphere may enable better constraints to be placed on this factor depending on the BC source being sampled, and thus improve the accuracy of future SP2 measurements of BC mass concentrations.

## 1 Introduction

Black carbon (BC) and brown carbon (BrC) aerosols are widely investigated components of atmospheric aerosol because they can absorb solar radiation and heat the atmosphere causing a positive radiative forcing of climate (Bond and Bergstrom, 2006;Liu et al., 2020a;Bond et al., 2013;Haywood and Shine, 1995). BC is emitted by incomplete combustion processes, including from anthropogenic (e.g., diesel engines) and natural (e.g., flaming combustion in wildfires) sources. BrC aerosols are organic aerosols that absorb light in the visible and near-UV regions, and are emitted directly from biomass burning, biofuel combustion and biogenic processes (Ramanathan et al., 2007;Bond, 2001;Andreae and Crutzen, 1997), or formed through the chemical reaction processes in the atmosphere, including the nitration of aromatic compounds (Zhang et al., 2013;Lu et al., 2011;Harrison et al., 2005;Lin et al., 2015), formation of higher molecular weight oligomers by acid catalysed aldol-condensation reactions (Shapiro et al., 2009;Bones et al., 2010;Noziere and Esteve, 2007), and reactions of ammonium-containing species with (di-)carbonyl species (Maxut et al., 2015;Powelson et al., 2014;De Haan et al., 2017). Typically, the light absorption coefficient for BC and BrC is

wavelength dependent over the visible spectrum, with BrC exhibiting a stronger wavelength dependence characterised by increasing absorption at progressively shorter visible wavelengths (Kirchstetter et al., 2004;Corbin et al., 2019;Voliotis et al., 2017).

Although BC and BrC are very important for climate, they are poorly represented in atmospheric models (Zuidema et al., 2016). This is in part due to the complex microphysical properties of BC and the lack of accurate refractive index (RI) descriptions for both BC and BrC (Liu et al., 2020b). Fresh soot particles often exist in the form of aggregates composed of primary spherules with an irregular and highly fractal geometry (Xiong and Friedlander,

2001;Wentzel et al., 2003). The morphology of these aggregates change markedly during the atmospheric aging process, influencing the corresponding particle size and optical properties (Zeng et al., 2019;Zhang et al., 2008). For example, after condensation of gaseous species such as sulfuric acid or water (under high relative humidity (RH) environments) on soot particles, or coagulation with the pre-existing particles, soot particles can experience

restructuring and the shape of the soot particles becomes more similar to a spherical particle (Zhang et al., 2008). The morphology of BC particles can be measured directly by using Scanning Electron Microscopy (SEM) or Transmission Electron Microscopy (TEM) (Fu et al., 2006;Chen et al., 2018;Ellis et al., 2016). However, the SEM/TEM approach only provides particle shape information in two dimensions and do not provide real time characterisation.

Alternatively, the particle morphology can be determined by measuring its size and mass with different techniques (Chen et al., 2018;DeCarlo et al., 2004). A conventional approach is to classify particles (generally using a differential mobility analyser, DMA, to select monodisperse particles on their mobility size) and then measure particle mass using a particle mass analyser (Zhang et al., 2008;Park et al., 2003;Park et al., 2004a;Park et al., 2004b;Chen

et al., 2018) (Wu et al., 2019). From the resulting information about particle mass for

different particle mobility sizes, the dynamic shape factor ($\chi$, defined as the ratio of the drag force on the particle divided by the drag force on the particle's volume equivalent sphere) and fractal dimensions ($D_f$) can be retrieved (DeCarlo et al., 2004). Mobility-mass fractal dimension ($D_{fm}$) has been reported over a wide range of 2.2-2.8 for diesel exhaust particles (Park et al., 2004b). $D_{fm}$ has been reported as higher than the $D_f$ - defined as the scaling exponents between the radius of gyration of an aggregate and the radius of primary spherules composing the aggregate - but the two are not always directly equivalent, particularly in the transition regime (Sorensen, 2011).

In recent decades, the RI of BC or BrC was derived by measuring the optical properties for particles of controlled size, with studies commonly utilising the DMA to classify charged aerosols on their electrical mobility diameter (Cotterell et al., 2020). However, the DMA approach suffers from transmitting a highly polydisperse and multi-modal distribution in terms of physical size. Since this technique relies on the selection of particles according to their electrical mobility from an aerosol ensemble with a distribution of charges, transmitting particles with a single electrical mobility can be achieved for multiple combinations of particle charge and size. Thus, in addition to singly-charged particles of the desired size, larger particles with charge states greater than unity are selected which can impact optical measurements significantly. Careful consideration of the impacts of multiply charged particles on subsequent RI derivations can go some way to reducing uncertainty in the resultant RI, but this nevertheless remains a significant contributor to uncertainty (Cotterell et al., 2020;Zarzana et al., 2014;Miles et al., 2011). Thus, the classification of particles without relying on electrical charge should reduce the uncertainty in refractive index retrievals from measured aerosol optical properties. Important additional considerations in the retrieval of refractive indices from optical spectroscopy data are the aerosol morphology (described

above) and mixing state. The mixing state can be probed using the Single Particle Soot Photometer (SP2), which can measure the refractory BC (rBC) mass content and optical size of individual particles. However the SP2 needs an empirical calibration to retrieve the rBC mass from the incandescence signal (Laborde et al., 2012a). The conventional method to calibrate the incandescence channel of SP2 is using size selected Aquadag standards (Acheson Inc. USA) and then correcting to a calibration representative of ambient rBC by a constant factor of 0.75 (Laborde et al., 2012b). However, few experiments since have independently verified this across various soot types.

To address the issues mentioned above, the Soot Aerodynamic Size Selection for Optical properties (SASSO) project utilised the Aerodynamic Aerosol Classifier (AAC) to classify particles according to aerodynamic diameter for size and mass distribution measurements and optical evaluation (Tavakoli et al., 2014). Specifically, SASSO has used the AAC size selection of emissions from wood burning, diesel combustion and secondary organic aerosol (SOA) formation, prior to optical measurements using cavity ring-down and photoacoustic spectroscopy with the EXtinction, SCattering and Absorption of Light for AirBorne Aerosol Research (EXSCALABAR) instrumentation, custom-built by the Met Office (Cotterell et al., 2020;Cotterell et al., 2019a;Davies et al., 2018).

The purpose of this paper is to determine the physical and chemical properties of BC and organic aerosols from different combustion sources representing the combustion of gas (methane), liquid (diesel) and solid (wood), and to examine the variation in calibration SP2 constants for BC particles generated by these combustion sources to enable accurate characterization of BC mass concentrations and mixing state in future studies. The key objectives of this work are:

(1) Develop reliable and repeatable methods of generating isolated BC and BrC using controlled combustion sources and a smog chamber

(2) Derive the dynamic shape factor of BC particles and material density for organic aerosols

(3) Determine mass spectral profiles of organic aerosols produced from wood combustion from an aerosol mass spectrometer (AMS)

(4) Explore the restructuring of BC particles in response to the controlled coating and

humidification of aerosol samples

(5) Investigate the response of the SP2 incandescence signal to different types of the BC particles.

The optical data presented in this paper is limited to the qualitative characterisation of the materials under investigation for the sake of achieving objective (1). The quantitative

parameterisation of the optical properties of the particles and associated development and application of ambient optical property models will be the subject of future publications.

## 2  Experimental setup and methods

Figure 1 shows our instrument configuration during SASSO. We used the AAC to classify aerosols based on their aerodynamic size prior to characterizing the particle size distribution,

chemical composition, aerosol mixing state and optical properties for the AAC-selected aerosols. The Centrifugal Particle Mass Analyzer (CPMA) and Scanning Mobility Particle Sizer (SMPS) sampled aerosols downstream of the AAC to measure the mass and mobility size distributions of the AAC-selected particles. A Nafion humidifier and a custom-designed thermal denuder (TD) were used for the BC restructuring test.

The right panel in Fig. 1 shows the various instrument configurations used in this study to
       target different measurements: Instrument set (1) was used for measurements of aerosol
       optical properties (and thereby enable the retrieval of refractive index for BC/BrC aerosols,
       subject of a future publication) and organic chemical measurement for wood combustion
       (setup (a)) and chamber (setup (c)) experiments. Instrument set (2) was used for dynamic
shape factor measurements for BC and material density measurement for organic aerosols
       (setup (a) and (b)). Instrument set (3) was used for SP2 incandescence signal calibration
       (setup (b)). Instruments set (4) was used for BC restructuring experiments (setup (c)).

## 2.1 Instrumentation

### 2.1.1 Aerodynamic Aerosol Classifier (AAC)

The AAC (Cambustion Ltd, Cambridge, UK) is used to select aerosols within a narrow range
       of aerodynamic diameters and does not suffer from the issue of multiple charges that affects
       selection using instruments such as the CPMA and DMA. The AAC uses a centrifugal force
       and sheath flow between two concentric rotating cylinders to produce an aerosol classified by
       aerodynamic diameter. The detailed information regarding the principle of AAC can be found
in Tavakoli et al. (2014).

### 2.1.2 EXtinction, SCattering and Absorption of Light for AirBorne Aerosol Research (EXSCALABAR)

The EXSCALABAR instrument used in this work was developed by the Met Office (Exeter, UK), which can be operated in both the laboratory and from the UK atmospheric research
aircraft (FAAM BAe-146). The operating principle of EXSCALABAR has been described in detail in previous papers (Davies et al., 2018;Cotterell et al., 2020;Cotterell et al., 2019a;Cotterell et al., 2019b). In brief, the instrument uses cavity ring-down spectroscopy

(CRDS) to measure the dry aerosol extinction at 405 and 658 nm wavelengths and photoacoustic spectroscopy (PAS) to measure the dry aerosol absorption coefficient at the wavelengths of 405, 514 and 658 nm wavelengths. For deployment during SASSO, all cells shared common sample conditioning; the aerosol sample relative humidity was reduced to <10% by passing through a Nafion drier (Perma Pure LLC) and Ozone and NOx were removed by an activated charcoal "honeycomb" scrubber (custom-built in-house). The sample passed through an impactor (Brechtel Manufacturing Inc., custom built for an 8 L min$^{-1}$ volumetric flow rate) with aerodynamic cut-off (D50) of 1.3 μm before being drawn through a series of flow splitters that evenly distributed the aerosol-laden air samples to the various optical spectrometers, each operating at 1 L min$^{-1}$. For both the 405 nm and 658 nm wavelengths, the PAS cells were mounted downstream of the CRDS cells (CRDS-PAS) i.e. the sample passed through the CRDS cell before entering the PAS cell of the same wavelength. All other cells operated in a parallel flow configuration. During the wood combustion experiments, a Condensation Particle Counter (CPC) (Model 3776, TSI, USA) was put in series after the 405-nm CRDS-PAS to measure the number concentration of particles contributing to the extinction and absorption signal. For an additional 405-nm CRDS spectrometer, the aerosol sample passed through a HEPA filter prior to sampling to provide a continuous measurement of the baseline and highlight the presence of any gaseous absorbers affecting measurements at that wavelength. CRDS and PAS measurements rely on characterization of the aerosol-free background. Before and after each wood combustion experiment (run), sample flow was routed through a HEPA filter immediately after EXSCALABAR's common sample inlet section to provide baseline measurements of empty-cavity ring-down time and background photoacoustic response for all CRDS and PAS spectrometers, respectively. During the Manchester chamber experiments, the CPC was fitted in parallel with the sample lines, and automated baseline measurements were made every 10

minutes. PAS cells with improved sensitivity – as described by Cotterell et al. (2019b) – were gradually implemented during 2019; only the 405 nm dry absorption measurement used the new cells during the Wildfire lab wood combustion experiments, but all cells were upgraded by the start of the Manchester aerosol chamber experiments. The PAS cells were calibrated using Ozone before and after each set of experiments as well as at least every working week during the experimental periods, as described by Davies et al. (2018) and discussed further by Cotterell et al. (2019a).

**2.1.3 Single Particle Soot Photometer (SP2)**

The refractory black carbon (rBC) mass concentration was measured by a SP2 (Droplet Measurement Technologies, Colorado, USA). The SP2 uses the laser-induced incandescence to measure the rBC mass and optical size of individual BC particles with an intra cavity Nd:YAG laser operating at 1064 nm. The particle size can be determined by detecting the laser signal scattered by particles, with the scattering intensity maximum related to the optical particle diameter through a calibration using polystyrene latex spheres. The optical size of BC-containing particles is determined by matching the measured scattering signal with calculations from light scattering calculations assuming a core-shell structure (core-shell Mie theory) (Moteki and Kondo, 2007). Because the scattering signal of the absorbing particle will be distorted during its transit through the laser beam due to the mass loss by laser heating, the leading edge scattering signal before the onset of volatilisation is extrapolated to reconstruct the scattering signal of the absorbing particles (Gao et al., 2007;Liu et al., 2014). This calibration was performed at the start of the measurement campaign. For those particles which contain absorbing materials such as the refractory BC, they will absorb 1064 nm light and then heat up and emit visible thermal radiation (incandesce). This incandescence signal is directly proportional to the mass of rBC as determined by a calibration (Liu et al., 2010) with

generated BC aerosols of known or independently-measured mass. In the atmosphere, besides BC, other materials (e.g., some metals and minerals) can incandesce at 1064 nm as well. However, as boiling point temperatures of these materials are rather different to that of black carbon, it is easy to distinguish them in measurements made using the SP2 equipped with an additional narrowband incandescence detector, such as the one used here (Liu et al., 2018). Recently, Sedlacek et al. (2018) demonstrated that charring of light-absorbing organic particles at 1064 nm can produce the refractory black carbon and then overestimate the rBC mass concentration, however this does not affect pure BC particles and we saw no evidence for an incandescence signal associated with the pure organic particles measured in this study. We investigated the effectiveness of this calibration from different types of BC particles in this study and report the outcomes of these investigations in Sect. 3.3.

### 2.1.4 High-Resolution Aerosol Mass Spectrometer (HR-AMS)

Non-refractory aerosol chemical compositions including sulfate, nitrate, ammonium, chloride, and organics were measured by a HR-AMS in real time. The HR-AMS was operated in fast-mode to capture the fast transition of the combustion phase during the wood combustion experiment (Kimmel et al., 2011). The instrument operation and data analysis of HR-AMS have been described in detail elsewhere (Alfarra et al., 2006; Allan et al., 2003, 2004; DeCarlo et al., 2006). The ionization efficiency of the AMS was calibrated using monodisperse ammonium nitrate particles according to the method described by Jayne et al. (2000) at various times during the experimental periods.

### 2.1.5 Centrifugal Particle Mass Analyzer (CPMA)

The CPMA (Cambustion Ltd., Cambridge, UK) uses opposing electrical and centrifugal fields to classify particles according to their mass-to-charge ratio. The ability to vary the

265 electrical field and rotation speed enables particle selection based on their mass. The principles and operation of the CPMA have been described elsewhere (Olfert and Collings, 2005;Olfert et al., 2006). Combining the CPMA with a CPC (Model 3776, TSI, USA) and scanning across the mass range of interest provides the bulk aerosol mass size distribution. As the CPMA uses an electrical classification method to select particles, it also suffers from

270 an issue of multiple charging similar to a DMA, and it has problems associated with a fraction of the uncharged particles that are also transmitted, particularly at the lower rotation speeds. In this study, an electrical ioniser (MSP Corp., USA) was used for wood combustion experiments and a $^{90}$Sr radioactive ioniser was used for chamber experiments to neutralize particles before they were sampled by the CPMA.

**2.1.6 Scanning Mobility Particle Sizer (SMPS)**

Aerosol size distributions in the diameter range from 14.9 to 673.2 nm were measured by a commercial SMPS (TSI, USA, employing a model 3082 classifier unit, 3081 DMA and 3786 CPC) with sheath and sample flow rates of 3 L min$^{-1}$ and 0.3 L min$^{-1}$, respectively. The DMA was operated in scanning mode with a scan time of 60 s and a retrace time of 4 s. The particle

size distribution was corrected for multiple charge effects and diffusion loss using the standard inversion algorithm in the SMPS software (AIM version 10). Before the experiment, the SMPS was calibrated using NIST certified polystyrene latex spheres (PSLs, Thermo Fisher Inc.).

**2.1.7 Transmission electron microscopy (TEM) sampling and analysis**

To investigate morphological properties, BC particles were collected on carbon coated copper grids using an electrostatic sampler (ESPnano, DASH Inc., USA) (Miller et al., 2010) for transmission electron microscopy (TEM) analysis. The grids were analyzed using a FEI Titan

operated at 300 kV. Several areas of the grid were imaged to provide representative images for each sample since both the grid film and particles contained carbon.

### 2.1.8 Thermal denuder (TD)

We explored the impacts of the partitioning of coating materials on the structure of BC. For this BC restructuring experiment, a thermal denuder was utilised to remove the coating materials on coated BC particles. The home-built TD consisted of a stainless steel tube in a temperature controlled furnace (Voliotis et al., 2021). The TD had a length of 0.97 m and an internal diameter (ID) of 0.15 m. Aerosols entered and exited the TD unit via a cylindrical 0.12 m long and 0.037 m ID stainless steel compartment. The temperature in the heating section (0.51 m × 0.15 m ID) was controlled by four PID controllers (Watlow EZ-ZONE) with additional temperature sensors on the outside of the tube. It is necessary to ensure flow through the TD whenever it is heated, even when bypassed to allow measurements of the unheated sample. A constant 2-2.5 L min$^{-1}$ flow of the sample air through it. A vacuum line maintained 2.0-2.5 L min$^{-1}$ through whichever of the bypass or TD line was not in use. The residence time of the air sample in the heating section was 216-270 s. The TD was given 30 mins for its temperature to stabilize before sampling. The temperature of the TD was calibrated by measuring the temperature at the axial centre of the denuder's heating zone. In this study, the purpose of the TD is to remove as much of the organic coatings from the combustion-generated particles as possible, rather than probing the volatility properties of the coating. Therefore, all heating zones of the TD were set to their maximum temperature of 180 °C. This upper temperature is lower than that achieved by other commercial TD units and minimises the risks of charring.

### 2.1.9 Nafion humidifier

The multi-tube Nafion humidifier in the Manchester home-made Hygroscopicity Tandem Differential Mobility Analyser (HTDMA) system was used for the BC restructuring experiment. The principle and configuration of the HTDMA system including the Nafion humidifier flow system can be found in Good et al. (2010). The RH in the Nafion humidifier was controlled by adjusting the relative mixing ratio of dry air from a compressed air source with humidified air generated by bubbling compressed air through a water-filled glass bulb.

**2.1.10 The Manchester Aerosol Chamber**

The Manchester Aerosol Chamber comprises an 18 m$^3$ collapsible Fluorinated ethylene propylene Teflon bag (3m (H) × 3m (L) × 2m (W)). It was used by Alfarra et al. (2012) to investigate the effect of photochemical aging and initial precursor concentration on the composition and hygroscopic properties of secondary organic aerosol. The chamber was run as a batch reactor in which the composition of the gaseous precursors, oxidising environment, primary emissions or seed particles, relative humidity and temperature were controlled. Air was supplied to the chamber by a blower at a flow of 3 m$^3$ min$^{-1}$. The air was dried and filtered for gaseous impurities and particles using a series of Purafil (Purafil Inc., USA), charcoal and HEPA filters (Donaldson Filtration, USA), prior to humidification with ultrapure deionised water. Halogen bulbs and two 6 kW Xenon arc lamps were mounted on the inside of the enclosure housing the bag, which was coated with a reflective space blanket (mylar) to maximise the irradiance in the bag and to ensure even illumination. The Xenon arc lamps were mounted on two opposite sides of the enclosure at different heights. The combination of illumination was tuned and evaluated to mimic the atmospheric actinic spectrum over the wavelength range 290 - 800 nm, and had a maximum total actinic flux of 1.4 × 10$^{14}$ (photon s$^{-1}$ m$^{-2}$ nm$^{-1}$) over the region 460 - 600 nm. The calculated $j(O^1D)$ value during the reported experiments was 1.23 × 10$^{-5}$ s$^{-1}$ (290 - 340 nm) and $j(NO_2)$ was 1.5 × 10$^{-3}$

s$^{-1}$ (290 - 422 nm). The relatively large volume of the chamber allows the dilute sample to be held for several hours without significant aerosol removal from wall losses, allowing the study of particles introduced directly or formed within the chamber over a period of several hours. The mean number and mass wall loss rates of particles inside the chamber were estimated as $9.17 \pm 1.3$ and $8.16 \pm 1.5 \times 10^{-5}$ s$^{-1}$, respectively (Shao et al., 2021). Further

details on the chamber are given by Alfarra et al. (2012).

**2.1.11 Diesel Engine**

The engine used in this study was a Volkswagen 1.9L SDI light duty diesel engine (EURO 4 car equivalent), mounted on a test rig (CM12; Armfield Ltd., Hampshire, UK) and coupled to an eddy current dynamometer. This setup was used for investigating the light absorption

properties of black carbon aerosol by Liu et al. (2017) previously. The engine throttle and the load on the dynamometer were controlled with dedicated software. The engine exhaust was passed through an oxidising catalytic converter (a retrofitted diesel oxidation catalyst consisting of a mixture of platinum and rhodium) followed by a standard Volkswagen silencer, then a computer operated pneumatic valve connected to the chamber, with 2 inch ID

stainless steel tubing between each component. The whole length of the exhaust line was 4 m. The fuel used was standard UK low sulphur diesel, obtained from a local fuel station.

For each run, the engine was started and warmed up at 2000 rpm with 30% load. Once the engine reached a steady temperature (after ~ 10 mins), a controlled amount of the exhaust was injected into the Manchester Aerosol Chamber by switching the valve for a set time

(referred this condition as "hot engine"). For the cold start runs (in which no throttle or load was applied, denoted as "cold idle"), the exhaust sample was injected within a minute of the engine starting.

### 2.1.12 Inverted flame burner

A miniature inverted soot generator (MISG, Argonaut Scientific) operated with propane was used to generate black carbon aerosol. The design, operation, and performance of the MISG has been previously reported (Kazemimanesh et al., 2019;Moallemi et al., 2019). The MISG was operated under two conditions: with the flow rate of propane and air at 0.0625 L min$^{-1}$ and 10 L min$^{-1}$ respectively (Condition 1), and 0.0625 L min$^{-1}$ and 7.5 L min$^{-1}$ respectively (Condition 2). Under these two conditions, the MISG produces black carbon with average Elemental Carbon/Total Carbon of 95%, determined via thermal-optical analysis.

### 2.2  Derivation of aerosol dynamic shape factor and material density

In this study, the SMPS and CPMA were arranged downstream of the AAC to measure the mobility diameter ($D_m$) and mass ($M_p$) of the particles within a narrow size range selected by the AAC based on their aerodynamic diameter ($D_a$). These parameters enabled the dynamic shape factor ($\chi$) of the BC particles to be derived by using Eqn. (1) under the assumption that the density ($\rho_p$) of pure BC is 1.8 g cm$^{-3}$ (DeCarlo et al. (2004)),

$$\chi = \frac{D_m C(D_{ve})}{D_{ve} C(D_m)} \qquad (1)$$

in which $C(D)$ is the Cunningham slip correction factor for a particle of diameter $D$ using the parameters specified by Kim et al. (2005), and $D_{ve}$ is volume equivalent diameter which can be calculated from $M_p$ using Eqn. (2).

$$D_{ve} = \left(\frac{6M_p}{\pi \rho_p}\right)^{\frac{1}{3}} \qquad (2)$$

$\chi$ can also be determined by using the specified $D_a$ in combination with the SMPS-measured $D_m$. Specifically, we can calculate $D_{ve}$ from $D_a$ for an initial value of $\chi_{trial}$ using Eqn. (3):

$$D_{ve} = D_a \sqrt{\chi \frac{\rho_0}{\rho_p} \frac{C(D_a)}{C(D_{ve})}} \qquad\qquad (3)$$

in which $\rho_0$ is the reference density (1 g cm$^{-3}$). This value $D_{ve}$ is then substituted into Eqn. (1) to calculate $\chi$. We then iterate through a range of $\chi_{\text{trial}}$ until $\chi$ and $\chi_{\text{trial}}$ converge. The equations were solved iteratively using a standard Brent's method solver ('findroots' command, Igor Pro version 6.36, Wavemetrics).

In addition, the material density ($\rho_m$) of organic aerosols can be calculated from Eqn. (4) and

(5) by assuming the organic particles are spherical (i.e., $\chi = 1$) (Adachi et al., 2019).

$$\rho_m = \left( \frac{6M_p}{\pi D_{ve}} \right)^{\frac{1}{3}} \qquad\qquad (4)$$

$$D_{ve} = D_m \quad \text{(when } \chi = 1 \text{)} \qquad (5)$$

## 2.3   Experimental methods

### 2.3.1 Wood combustion experiment

Our wood combustion experiments were designed to expand on the studies reported by Haslett et al. (2018) to produce repeatable combustion events with discernible transitions between the three burning phases of pyrolysis, flaming, and smouldering combustion. Haslett et al. (2018) used the FM Global Fire Propagation Apparatus (FPA) (FTT, East Grinstead, UK) as their controlled ignition source while we used the iCone Calorimeter (Fire Testing

Technology, FTT, East Grinstead, UK) located in the Wildfire laboratory at the University of Exeter (Fig. 1(a)). The main difference between the FPA and iCone approaches is that the there is a forced flow of air from beneath the sample in the FPA, whereas in the iCone the sample sits in ambient air conditions. Otherwise, both approaches rely on oxygen consumption calorimetry to measure the heat release rate from a burning object.

The aim of the wood combustion experiments was to produce different phases of combustion that were as well separated as possible, rather than under natural burning conditions. This allowed the aerosol emissions from different burn phases: pyrolysis, flaming and smouldering, to be analysed separately. Three core wood types were selected for all the experiments which we hypothesised would produce different burn conditions, i.e. different durations of pyrolysis,

flaming and smouldering. These wood types were: *Sequioadendron gigantum* (Giant Redwood), *Pinus sylvestris* (Scots Pine), and *Populus nigra* (Poplar) with bulk densities of 0.45, 0.51, $0.4 \times 10^3$ kg/m$^3$ respectively. The Giant Redwood and Scots Pine are gymnosperm 'softwoods', with the former considered to have low density and the latter a high density, while the Poplar a low density angiosperm 'hard wood' species. In addition we also sampled

combustion emissions from *Thuja plicata* (Western Red Cedar) with a bulk density of $0.38 \times 10^3$ kg/m$^3$. Both Scots pine and Western red cedar are resin rich, while the Giant Redwood contains less resin, and the Poplar contains barely any resinous compounds. Hence, all should be capable of producing different volumes and types of aerosol particles. There are many factors that can influence composition and properties of generated aerosols during

combustion, such as the wood resin content and moisture content, which would result in a highly extensive variable set. In the work presented here, we only focus on the influence of water content of wood samples on the physical and chemical properties of the particles from wood combustion. We assessed wood of two different fuel moistures: fully oven dried samples and moist samples with ~25% moisture content.

During the experiment, a wood sample (L×W×H (mm): 90×90×30) was first placed into a custom-made stainless steel basket and then exposed to a radiant heat flux of 40 kW m$^{-2}$ using the iCone Calorimeter. This radiant heating thermally decomposes the wood sample and generates volatile gases (pyrolysate) as the wood begins to pyrolyse. A continuously

operated spark igniter that was positioned in the released stream of pyrolysate acted as a

source of ignition. Immediately following the initial placement of the wood sample under the

radiant heat source, pyrolysate was continuously released and the pyrolysate concentration

immediately above the wood sample and in the vicinity of the spark igniter increased also.

The sample ignited once this pyrolysate reached sufficient concentrations and was well mixed

with the surrounding air, at which point the igniter was switched off and removed from the

air flow. Measurements were taken as soon as a sample was exposed to the radiant heat flux.

Each sample was allowed to flame for 5 minutes and the aerosols captured from this phase

and then the flames were snuffed out manually leaving the fuel smouldering so that separate

measurements of the smouldering phase could be taking alone for a further 5 minutes.

Throughout the experiment, the exhaust emissions from combustion were collected in a hood

and entered into an exhaust duct. The concentration of oxygen, carbon dioxide and carbon

monoxide in the exhaust gas were measured by gas analysers integrated into the iCone

system and used to calculate the heat release rate from the burning fuel. We installed an

outshoot on this duct to allow part of the exhaust to be carried to a separate sampling system

that allowed the measurement of the aerosol properties in the exhaust. Thus monitoring could

occur in real time by the instruments (described above and shown in Fig. 1), after the exhaust

was diluted with the compressed air through a set of Dekati DI-1000 ejector diluters.

**2.3.2 SP2 incandescence signal calibration and BC morphology investigation experiment**

For the purposes of SP2 incandescence signal calibration and morphology investigation of

BC particles, three types of the BC particles other than that emitted from wood combustion

were investigated through chamber experiments (Fig. 1(b)). For the BC particles generated by

the MISG: BC particles were introduced into the chamber through a Dekati DI-1000 ejector

diluter. For the BC particles from the Aquadag standards: the Aquadag standards was first

generated by an aerosol atomizer (Topas ATM 226) and then passed through a diffusion drier (using silica desiccant) prior to entering the chamber. For the BC particles emitted from the engine: a hot engine running condition (2,000 r.p.m., 30% load and 10 min warm up) and a cold idle condition were investigated. After the BC mass concentration had accumulated to adequate levels in the chamber, a catalytic stripper (operated at 350°C, Model CS10, Catalytic Instruments, Germany) was used to remove the coatings on the BC particles before they were sampled by the instruments.

### 2.3.3 SOA formation through chamber experiments

For the purpose of testing models of BC and SOA mixing during the SASSO project, two procedures for generating SOA were used: one that yielded non-absorbing SOA and one that produced brown carbon. Experiments were conducted in the photochemical aerosol reaction chamber at the University of Manchester. During the experiment, $NO_2$ (10% v/v, with a balance gas of high purity $N_2$ (BOC, UK)) was injected directly into the bag from a custom-made gas cylinder via stainless steel tubing, and its concentration was measured using a chemi-luminescence gas analyzer (Model 42i, Thermo Scientific, MA, USA). After the $NO_x$ concentration in the bag reached the desired value, the precursor VOC was injected into a heated glass bulb (80°C) and flushed into the bag with high purity nitrogen. Subsequently, chamber lights were turned on to trigger SOA formation. The EXSCALABAR system was used to determine the light absorption properties of the formed SOA particles in real time throughout the experiment.

In this study, 50 ppb $NO_x$ and 250 ppb α-pinene (Sigma-Aldrich) was used to form non-absorbing SOA; this route to forming non-absorbing SOA is well characterized and has been reported previously (Nakayama et al., 2010). Here, we only focus on the methodology used to produce the brown carbon SOA. Previous work has reported 'brown' SOA formation

through oxidation of aromatic precursors under high $NO_x$ conditions (Laskin et al., 2015). In this study, we injected ~ 400 ppb cresol (Sigma-Aldrich) and ~ 15 ppb $NO_2$ (~ 60 ppb $NO_x$ in the chamber) into the chamber, and removed the UV filter from one of the Xenon arc lamps to increase photochemistry and accelerate and enhance the 'brown' SOA formation.

### 2.3.4 BC restructuring experiment

For the purpose of testing the influence of organic coatings and relative humidity on the structure of the BC particles, a series of experiments were conducted in this study. First, bare BC particles with minimal accompanying VOCs were injected into the chamber by using an oxidising catalytic converter, heated ejector dilutors, a catalytic stripper, Purafil, and activated charcoal denuders between the exhaust line of the engine and the chamber inlet. After the BC reached the desired concentration, BC injection was stopped. The particle size distribution of the dried bare BC particles was scanned by the AAC. In addition, the particle size distribution of the dried bare BC particles following 'humidity cycling', i.e. exposure to 90% RH for around 10 s and then dehydration to 10% RH, was measured by AAC for comparison. Afterwards, we injected 50 ppb $NO_x$ and 250 ppb α-pinene into the chamber. Photochemical reaction was initiated by turning the chamber lights on, and organic materials condense onto the BC particles. After the condensed organics equilibrate with the surrounding VOCs and the particles stabilised at a certain size (295 nm in aerodynamic size), the aerodynamic particle size distribution of the dried organic coated BC particles, and that of the dried organic coated BC particles experienced the humidity cycling, was measured by AAC. Hereafter, a thermal denuder operated at 180 °C was added, and the organic coatings of the coated BC particles, that had either experienced the humidity cycling process or not, were removed by the TD, and then the size distribution of the BC core was measured by the AAC and SMPS.

## 3 Results and Discussion

### 3.1 Evolution of BC and organics through wood combustion process

Fig. 2 shows an example time series of the rBC and organic aerosol (OA) emitted during combustion for a representative wood sample. Throughout the wood combustion experiment, noticeable changes in BC and OA concentrations in the exhaust gases were observed. Following exposure of a wood sample to the heat flux of the iCone and prior to ignition, a clear signal corresponding to the production of organic aerosols was observed. For these *pyrolysis phase* aerosols, the ratio of the OA mass ($M_{OA}$, measured by the AMS) to the sum of masses attributed to OA and rBC ($M_{rBC}$, measured by SP2, $r_{OA}=M_{OA}/(M_{OA}+M_{rBC})$) was ~ 1.0, implying that aerosols produced during the *pyrolysis phase* of combustion were composed dominantly of organic species. This organic material must be due to the early stages of pyrolysis where compounds of cellulose and hemicellulose within the wood are being broken down during its thermal decomposition and the surface of the wood begins to char. We suggest that these pyrolysis phase aerosols are formed from the condensation of volatile gases into the condensed phase as the aerosol plume lofts upwards away from the heat source of the iCone and into the cooler environment of the exhaust system.

Immediately after ignition, the number (and mass) concentration of the OA decreased abruptly, while the rBC mass concentration increased sharply to 150 $\mu g/m^3$ (Fig. 2). We refer to this combustion phase following ignition as the *flaming phase*. The $r_{OA}$ at the onset of flaming combustion (the first 90s after ignition) is less than 0.05, indicating that most of the emitted aerosols during this period are BC. As flaming combustion progressed, the extent and depth of charring increased and as the flammable gas flux slows the flame begins to diminish. Char is a strong insulator and hence the char layer forms an obstacle to the conduction of heat into the lower uncharred layers of wood, reducing the production rate of the flammable gases

and decreasing the rate of heat release from combustion. As flaming subsides the number (and mass) concentration of BC particles decreases (Fig. 2) and the $r_{OA}$ increased to around 0.2 after 300 s after the ignition.

After ~300 s (5 min) from the time of ignition the flames were snuffed manually to force the combustion from the flaming to the smouldering phase, thus transitioning the oxidation

reaction from that on gas phase species to direct oxidation of the solid fuel. Fig. 2 shows that the number (and mass) concentration of rBC in the *smouldering phase* decreased abruptly as flaming combustion ceased and the OA concentration increased sharply. The $r_{OA}$ was 1.0 during this period, indicating that almost all the emitted aerosols are composed of organic species only.

**3.2 Brown carbon formation**

As shown in Fig. 3, after the precursors (~ 400 ppb cresol  and ~ 60 ppb $NO_x$) were introduced into the chamber and the lights were switched on, the SOA started to form and the particle size increased to ~ 200 nm after 40 minutes. Over this same time period, we measured increasing aerosol absorption and extinction coefficients, the single scattering

albedo (SSA) was observed to increase sharply at the onset of the SOA formation and then stabilized at around  0.81 (at 405 nm) eventually. Most of the evolution in SSA (i.e. increasing over time) is expected to be caused by the increase in particle size over a size range where SSA is very sensitive to particle size. We emphasise that we did not inject any primary ozone in this experiment, and the abrupt increase in apparent ozone concentration

around the time of VOC injection is likely caused by the cresol (as an aromatic compound) causing optical interference in the ozone instrument.

**3.3 SP2 incandescence signal calibration with different types of BC particles**

The most common way to calibrate the incandescence channel of SP2 is using monodisperse Aquadag standards (Acheson Inc. USA), which is then corrected for ambient rBC by applying a constant factor of 0.75 based on a previous laboratory comparison (Laborde et al., 2012b). As the incandescence signal is not independent for the different types of BC particles, using a constant factor of 0.75 may represent an uncertainty in the retrieved BC mass concentration. Thus, quantifying the correction factor for many types of BC particles found commonly in the atmosphere may enable better constraints to be placed on this factor depending on the BC source being sampled, thus improving the accuracy of future SP2 measurements of rBC mass concentrations. The broad range of BC aerosols generated in this study from different combustion sources served as an ideal platform to assess the variation in this correction factor to SP2-derived rBC mass.

In this study, the incandescence signal of the SP2 was measured for BC particles from catalytically stripped diesel engine exhaust emissions, an inverted flame burner, and controlled flaming wood combustion, respectively, and compared with that measured from an Aquadag standard. The uncertainties here refer to precision of the fitted parameters reported by the Igor Pro fitting algorithm, based on analysis of residual data. As shown in Fig. 4, for the BC particles emitted from the diesel engine under hot engine and cold idle conditions (Fig. 4(a)), the slopes of the incandescence signal with BC mass are $0.821 \pm 0.002$ and $0.794 \pm 0.005$ times of that measured from the Aquadag standard, respectively. Note that while some deviation from a perfect linear response is noted, this is small compared to the variation in slopes, so represents a minor source of uncertainty in comparison. These correction factors are 9.4% and 5.6% different with the common value of 0.75 (with the uncertainty less than 5%) recommended by Laborde et al. (2012b) when deriving the mass concentration of BC emitted from diesel engines. For the BC particles generated from the flame burner (Fig. 4(b)),

the correction factor is $0.879 \pm 0.003$. Meanwhile, for the BC particles emitted from the flaming phase during the combustion of Scots pine, Poplar, Giant Redwood or Western red cedar, the correction factors are $0.913 \pm 0.009$, $0.906 \pm 0.014$, $0.889 \pm 0.027$ or $0.843 \pm 0.028$, respectively. We stress that, for the SP2 calibrations here from wood combustion emissions, the BC particles were not treated with a catalytic stripper before sampling by the SP2. While coating materials may char under 1064 nm to produce refractory black carbon and therefore cause overestimates in the incandescence signal (Sedlacek et al., 2018), as shown in Fig. 2(c), the BC particles generated at the beginning of the flaming phase contained almost no organic species, with $r_{OA}$ values less than 0.05. Even if this OC were to be converted to EC with 100% efficiency (which we consider to be highly unlikely), this would represent a very small error.

The differences in the correction factors derived in this study with the default value of 0.75 are 9.4% (5.6%), 17.2% and 12.4-21.7% for the BC particles emitted from engine with hot engine (or cold idle) condition, flame burner and wood combustion, respectively. We recommend that future studies utilizing the SP2 for rBC mass concentration measurements use SP2 calibrations with the same type of BC as that to be studied.

**3.4 Physical properties of black carbon and organic aerosols**

The morphology of BC particles change markedly during atmospheric aging, with associated impacts on the particle size and optical properties (Zeng et al., 2019;Zhang et al., 2008;Teoh et al., 2019). In this study, the dynamic shape factor of BC particles and the material density of organic aerosols were derived by measuring the mass and/or mobility diameter of AAC-selected particles. For biomass burning aerosols generated through our controlled combustion of dry wood samples, the AAC was set to pass aerosols with an aerodynamic diameter $D_a$=200 nm and we sampled BC or organic aerosols produced from the flaming or smouldering phases, respectively. Number concentration measurements from the CPC were

recorded as the CPMA was scanned to determine the aerosol mass distribution. Figure 5(a) shows that there are considerable differences in the mass distributions for the produced BC (during the flaming phase) and organic aerosols (during the smouldering phase). The peak mass of BC particles produced from the wood samples of the Scots pine, Poplar and Giant Redwood are 10.86, 12.47 and 7.00 fg, respectively. Meanwhile, the corresponding peak mass for the organic aerosols are 1.72, 1.74 and 1.90 fg, respectively. This difference is likely due to the difference in density and morphology for BC and organic aerosols. BC normally has a large density and a more irregular morphology than organic aerosols. After the peak masses for the AAC-selected aerosols were ascertained, the CPMA was set to these values to further select the AAC-selected aerosols ($D_a$=200 nm) by their peak mass for a subsequent mobility size measurement using our SMPS. Figure 5(b) shows that the corresponding peak mobility size of the BC and organic particles after selection according to aerodynamic diameter are 394, 405, 292 nm and 139, 139, 140 nm for Scots pine, Poplar and Giant Redwood, respectively. From these aforementioned peak mobility diameters and aerosol masses, the $\chi$ of the BC particles and the $\rho_m$ of the organics can be calculated from the methods described in Sect. 2.2. We highlight the discussion in Sect. 2.2 that stated that $\chi$ could be calculated using either the aerodynamic or volume-equivalent diameter; as organic species may evaporate during the AAC selection (due to the high sheath flow rate) and thus bias the $D_a$, only the mass and mobility size distributions were used in these calculations. Table 1 summarises the $\chi$ values inferred for the $D_a$=200 nm BC particles, with values of 2.17 ± 0.04, 2.10 ± 0.08 and 1.80 ± 0.12 ascertained for oven dried samples of Scots pine, Poplar and Giant Redwood, respectively. The $\rho_m$ of the organics are 1.22 ± 0.01, 1.23 ± 0.01 and 1.32 ± 0.03 g/cm$^3$ for oven dried samples of Scots pine, Poplar and Giant Redwood, respectively. For combustion experiments using wet wood samples (with moisture contents around 25%), the CPMA instrument was unavailable. Instead, the mobility size distributions

for the AAC-selected $D_a$=200 nm BC particles or organic particles were measured by the SMPS immediately after AAC-selection. As shown in Fig. 6, the peak mobility size of the BC and organic particles are 296, 256, 161 nm and 149, 142, 137 nm for Scots pine, Poplar and Giant Redwood, respectively. Based on their aerodynamic and mobility size, the χ of the

BC particles and the $\rho_m$ of the organics were calculated and summarised in Table 1. The χ of the $D_a$=200 nm BC particles are 1.85 ± 0.03, 1.67 ± 0.03 and 1.20 ± 0.02 for Scots pine, Poplar and Giant Redwood, respectively, lower than that of the dry wood samples. The $\rho_m$ of the organics are 1.44 ± 0.02, 1.52 ± 0.01 and 1.60 ± 0.01 g/cm$^3$ for Scots pine, Poplar and Giant Redwood, respectively, higher than that of the dry wood samples. This result implies

that the water contents of wood samples are important in determining the physical properties of emitted particles from their combustion, in addition to any influence on the pre-ignition pyrolysis phase. The densities of organic aerosols reported/used in previous studies vary significantly, from 0.6 to 1.4 g cm$^{-3}$ (Turpin and Lim, 2001;Nakao et al., 2013;Li et al., 2016). For the biomass burning aerosols, Zhai et al. (2017) reported the effective density ranged

from 1.35 to 1.51 g cm$^{-3}$ for the carbonaceous aerosol produced from the agricultural residue burning. Our results would expand the available data on the variation densities for organic aerosols from the wood combustion.

For the BC particles from the catalytically stripped Aquadag standard, diesel engine exhaust, and inverted flame burner, particles were first selected by the AAC prior to mobility size

distribution measurements using the SMPS. The χ of the BC particles were derived from the controlled aerodynamic size and the measured peak mobility diameter. Fig. 7(a) shows that the particle mobility diameter increases with the selected aerodynamic size for all three types of BC particles, while Fig. 7(b) shows the corresponding variations in χ. These figures demonstrate clearly that the dynamic shape factor varies with both the aerosol aerodynamic

size and with the BC source. For the BC particles generated from the Aquadag standard, the $\chi$

increased sharply from $1.315 \pm 0.004$ to $1.460 \pm 0.003$ as $D_a$ increased from 75 nm to 125 nm,

but then decreased with further increases in $D_a$ to a value of $1.255 \pm 0.005$ at $D_a$=500 nm.

This result implies that the formed BC particles are more irregular as $D_a$ increases initially,

but particles adopt more collapsed morphologies as $D_a$ increases further. These trends in $\chi$

likely arise from the coagulation of the primary BC particles, which drives the fractal-like

morphology for small BC particles and relative compact shape for large particles. For BC

particles generated from the diesel engine, the $\chi$ exhibits a monotonically decreasing trend

with particle size over the reduced size range probed for this BC source, with $\chi$ decreasing

from $1.329 \pm 0.003$ at $D_a$=150 nm to $1.243 \pm 0.002$ at $D_a$=275 nm. The TEM images (Fig.

7(c)) show that the primary spherules generated directly from the diesel engine are around 20

nm. As the BC particles selected in the experiment are much larger than the primary

spherules, the coagulation process drives the large particles towards more compacted

morphologies than the small particles and this leads to the decreasing trend of $\chi$ with particle

size. For the BC particles generated from the flame burner, the $\chi$ increases with particle size

for both operating conditions and a maximum value was reached of $2.731 \pm 0.021$ for $D_a =$

150 nm. The same increasing trend was reported by Slowik et al. (2004) previously, who

reported an increase in $\chi$ from ~ 1.3 to ~ 3.0 when the mobility size increased from 100 nm to

300 nm for the BC particles generated with lower propane/$O_2$ ratio compared to the

measurements we report here. The TEM images (Fig. 7(d)) shows that, similar to the BC

particles emitted from diesel engine, the primary spherules generated from the flame burner

are also around 20 nm. As our experiments on BC particles from the flame burner selected

particles at small sizes ($D_a$ in the range ~ $50 - 150$ nm), the increase of the dynamic shape

factor with $D_a$ is probably due to the coagulation of primary generated particles. In addition,

by comparing Fig. 7(c) and Fig. 7(d) (more images are shown in Fig. S1), it is clear that the large BC particles generated from the flame burner tend towards aggregate morphology, in contrast to the compact morphology of the large BC particles emitted from the diesel engine (Fig. 9(c)). This leads to the flame burner generated BC particles exhibiting larger $\chi$ than those from the diesel engine.

**3.5 BC restructuring**

Recent work has shown that soot particles maintain their structure after coating with SOA and subsequent thermal denuding (Bhandari et al., 2017), but will restructure in response to humidification (Leung et al., 2017). We performed a series of experiments to better understand this restructuring. As shown in Fig. 8, there is no clear change in the measured particle aerodynamic diameter for the bare BC after they experience the humidity cycling process, implying that the bare BC particles retain their structure even after they experience very large perturbations in RH, this result is similar to that reported by Bhandari et al. (2019). In addition, no size change of the organic coated BC particles was observed after they experienced the humidity cycling process either. Based on these results, we cannot form conclusions on whether the BC core was restructured during the humidity cycling process. There are two possibilities: (1) The BC cores retained their structure throughout humidity cycling process; or (2) As shown in Figure 8, the coatings on BC particles are very thick and therefore dominate the particle, rendering our size measurement approach insensitive to any changes in BC core size from restructuring. The size of the bare BC particles is around 68 nm, but after coating by SOA, the size of the coated BC particles reaches up to around 300 nm. Due to the very large coating thicknesses of the coated BC particles, even if the BC cores were restructured during the humidity cycling process, any changes were not reflected in the overall particle size as this was dominated by the contribution from the coating organics. To

further examine these possibilities, the organic coatings of the coated BC particles, that had either experienced the humidity cycling process or not, were removed by the TD, and then the size distribution of the BC core was measured by the AAC and SMPS. As shown in Fig. 9, a clear difference in BC core size was observed between the organic coated BC particles that had passed immediately through the TD (Fig. 9(a)) and those that had first experienced the humidity cycling process (Fig. 9(b)). The mobility diameter of the BC core size peaked at 79.3 nm for the former case but peaked at 23.6 nm for the latter one, implying that the BC core was restructured (becomes more compact) after coating with organics and then experiencing the high humidity environment.

**3.6 AMS Mass spectra of organic aerosols**

Owing to the complex nature of biomass combustion in the natural environment, and the high sensitivity to the combustion environment, the organic mass spectral signatures from previous AMS experiments vary considerably. This variability makes it difficult to estimate the contribution of biomass burning aerosol (BBA) to total PM through positive matrix factorisation (PMF) methods (Paglione et al., 2020). The mass spectra for the BBA generated from our repeatable and controlled laboratory experiments have the potential to tighten constraints on the AMS spectral signature(s) for BBA. Highly controlled and reproducible measurements of aerosol emissions from combustion of a common African biofuel source has been achieved and reported by Haslett et al. (2018) previously. Here, we expand detail of the mass spectra observations reported by Haslett et al. (2018) by providing data sets on the combustion for different types of biomass to that studied in previous work.

The mass spectra of organic aerosols generated from the pyrolysis and smouldering phases (noise to signal ratio is less than 0.005), and the difference mass spectra between these two phases, are shown in Fig. 10 and Fig. 11 for the combustion of dry and wet wood samples,

respectively. For all experiments, irrespective of the wood type and moisture content, the spectra for the pyrolysis and smouldering phases are dominated by hydrocarbon ion fragments, including $C_nH_{2n-1}$ (m/z=41, 55), $C_nH_{2n+1}$ (m/z=29, 43, 57), $C_nH_{2n-3}$ (m/z=67) and $C_{5+n}H_{5+2n}$ (m/z=77, 91). These peaks are associated with fragments of saturated alkenes, alkanes, cycloalkanes and aromatic compounds, respectively. Our results are similar to the mass spectra reported by Haslett et al. (2018) for the controlled combustion of biomass in the laboratory. By comparing Fig. 10 and Fig. 11, no clear difference in mass spectra of the organic aerosols in individual burn phases (pyrolysis or smouldering phase) with the moisture content of the wood samples are observed. This implies that the effect moisture has on the organic chemical profile of wood burning emissions is through changing the durations of the different phases of the burn cycle, not through the chemical modification of the individual phases. This is in stark contrast to the dominant role of the wood water content in determining the morphological parameters and densities of generated biomass burning aerosols shown in Sect. 3.2.

The peaks at m/z 57, 60 and 73 are seen together in both combustion phases; these peaks are often used as markers for biomass burning organic aerosols (Alfarra et al., 2007), as very few other aerosol sources in the natural environment can contribute to these peaks. Interestingly, for the wood samples of Scots pine, Giant Redwood and Western red cedar, these peaks at m/z 57, 60 and 73 were more dominant in the smouldering phase. This is a similar observation to that reported by Haslett et al. (2018). However, for the Poplar wood sample, the peaks are most dominant in the pyrolysis phase. This is particularly interesting because it suggests that the combustion of angiosperm wood (Poplar) and gymnosperm wood (Scots Pine, Giant Redwood and Western red cedar) will emit different chemical species depending on the combustion phase. This difference may be due to the resin content and composition

(which is related to chemical volatility) within the wood. Future work should consider the impact of burning of different wood types and their respective relationship to combustion phase.

For all wood samples, the peak at m/z 44, which is mainly attributed to the $CO_2^+$ fragment with a possible contribution from $C_2H_4O^+$, is more prominent in pyrolysis phase than the smouldering phase, which is in contrast with the results from Haslett et al. (2018). This could be a result of the different wood types. We note that the study of Haslett et al., (2018) used Rubber wood (*Hevea brasiliensis*), that produces latex sap. This hard tropical wood also has a bulk density much higher than the woods tested here (0.63 x $10^3$ kg/m$^3$) and likely a higher lignin content where lignin is one the last compounds to be broken down by the combustion process.

**4 Conclusions**

During the Soot Aerodynamic Size Selection for Optical properties (SASSO) project and a EUROCHAMP transnational access project, the physical and chemical properties of black carbon and organic matter from different combustion sources were investigated. For BC particles from the diesel engine and flame burner emissions, the TEM images show that the primary spherules size are similar, around 20 nm. As particle size increases, BC particles adopted more collapsed/compacted morphologies from the diesel engine but tend to more aggregated morphologies from the flame burner source. For the particles emitted from the wood combustion, the χ of BC particles and the $\rho_m$ of organics were observed to range from 1.8-2.17 and 1.22-1.32 g/cm$^3$ for dry wood samples, 1.2-1.85 and 1.44-1.60 g/cm$^3$ for wet wood samples. No clear difference in the AMS mass spectra of the organic aerosols in individual burn phases (pyrolysis or smouldering phase) with the moisture content of the wood samples was observed. This implies that the effect moisture has on the organic

chemical profile of wood burning emissions is through changing the durations of the different phases of the burn cycle, not through the chemical modification of the individual phases. In addition, the incandescence signal of SP2 was calibrated with the different types of BC particles generated in our study and compared that with the Aquadag standard. The correction factor, which is used for converting the incandescence signal from the Aquadag standard to the investigated BC, was measured as $0.821 \pm 0.002$ (or $0.794 \pm 0.005$), $0.879 \pm 0.003$ and $0.843 \pm 0.028$ to $0.913 \pm 0.009$ for the BC particles emitted from engine under hot running (or cold idle) condition, flame burner and wood combustion, respectively. These values are 9.4% (5.6%), 17.2% and 12.4-21.7% different with the default value of 0.75 used recently. Quantifying the correction factor for many types of BC particles found commonly in the atmosphere may enable better constraints to be placed on this factor depending on the BC source being sampled, and thus improve the accuracy of future SP2 measurements of rBC mass concentrations.

**Acknowledgements**

This work was supported by the UK Natural Environment Research Council (NERC) (grant ref. NE/S00212X/1) and the Met Office, and received a trans-national activity funding from the European Union's Horizon 2020 research and innovation programme through the EUROCHAMP-2020 Infrastructure Activity under grant agreement No. 730997. Prem Lobo was supported by the UK National Centre for Atmospheric Science (NCAS) Visiting Scientist Programme. Thanks to Catalytic Instruments for the loan of the catalytic stripper CS10, and to Xioamei Du  at National Research Council Canada for the TEM images.

**Author contributions.** D.A., D.H., R.A. D.L. J.H. and H.C. designed research; D.H., C.Y., R.A., K.S., J.L., M.C., C.B., I.R. and Z.L. performed wood combustion experiments; D.H., R.A., Y.S., M.D. and A.V. performed Manchester aerosol chamber experiments; B.S., P.L.

and G.S. performed the burner experiments. D.H. conducted the data analysis and wrote the

manuscript with inputs from all co-authors.

**Data availability.**

Processed data are available at https://doi.org/10.6084/m9.figshare.15172572.v1. Raw data is archived at the University of Manchester and is available on request.

**Competing financial interests**

The authors declare no competing financial interests.

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

# Figures and Captions

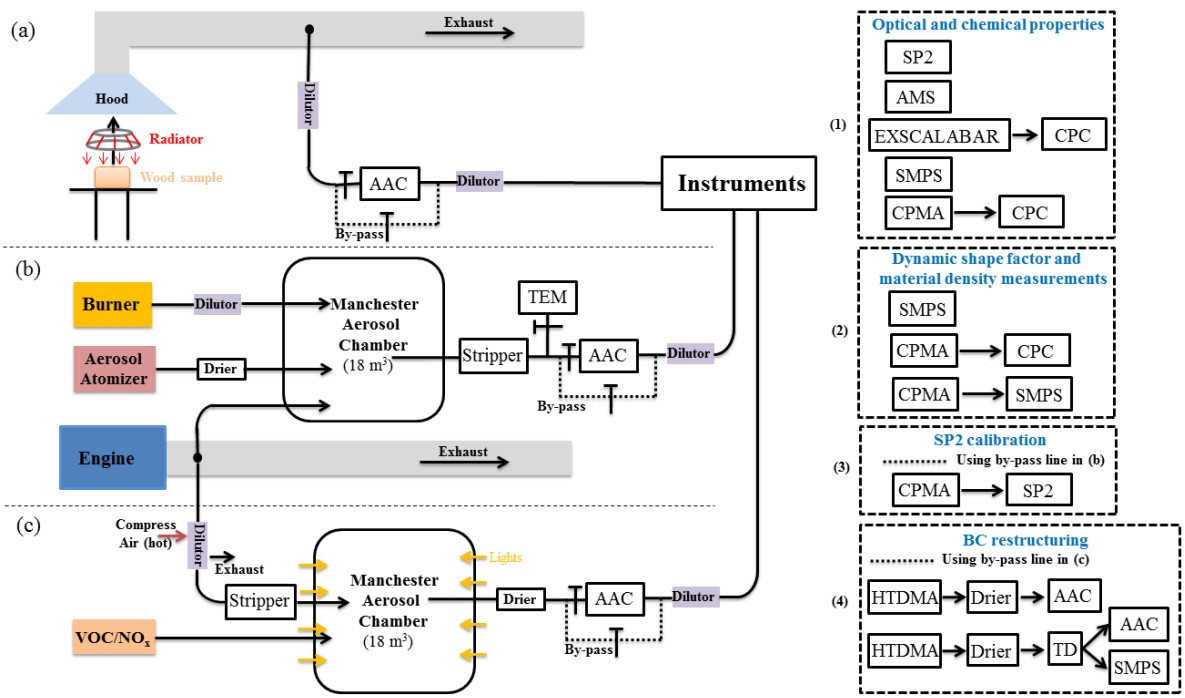

**Fig. 1.** Schematic diagram of the experimental configuration for: (a) Wood combustion; (b) SP2 incandesces signal calibration and BC morphology investigation; (c) Brown carbon formation and restructuring of BC. The combinations of instruments described in (1), (2), (3) and (4) represent different measurement configurations used to enable characterisation of specific aerosol physiochemical parameters, as described in the main text. The instrument set (1) was used for measurements of aerosol optical properties (and thereby enable the retrieval of refractive index for BC/BrC aerosols, subject of a future publication) and organic chemical measurement for wood combustion (setup (a)) and chamber (setup (c)) experiments; instrument set (2) was used for dynamic shape factor measurements for BC and material density measurement for organic aerosols (setup (a) and (b)); the instrument set (3) was used for SP2 incandescence signal calibration (setup (b)); the instruments set (4) was used for BC restructuring experiments (setup (c)).

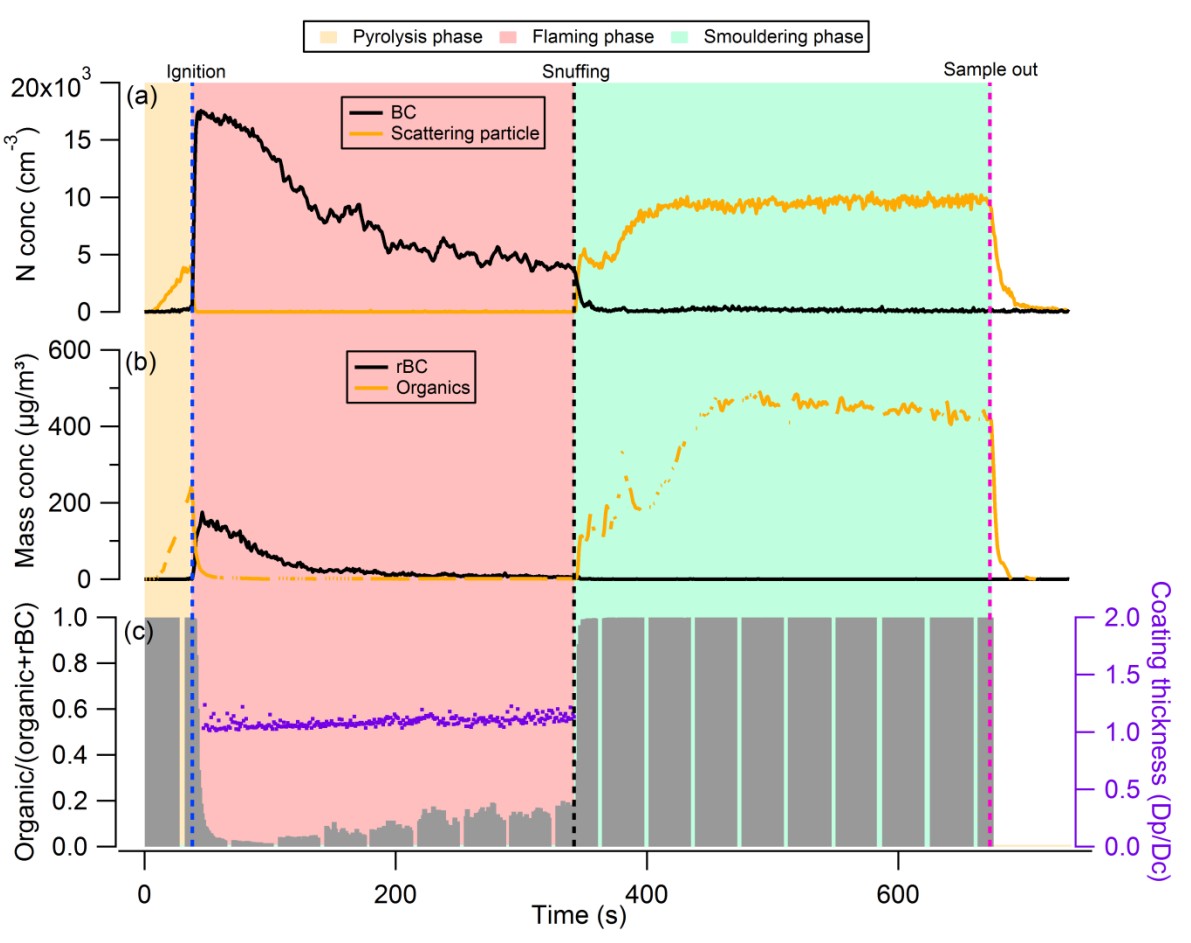

**Fig. 2.** Example of the phase transitions between different burn phases of dry Giant Redwood: (a) number concentration of black carbon and the scattering particles detected from SP2; (b) mass concentration of rBC and organics measured by SP2 and AMS, respectively; and (c) mass ratio of organic aerosol to total aerosol.

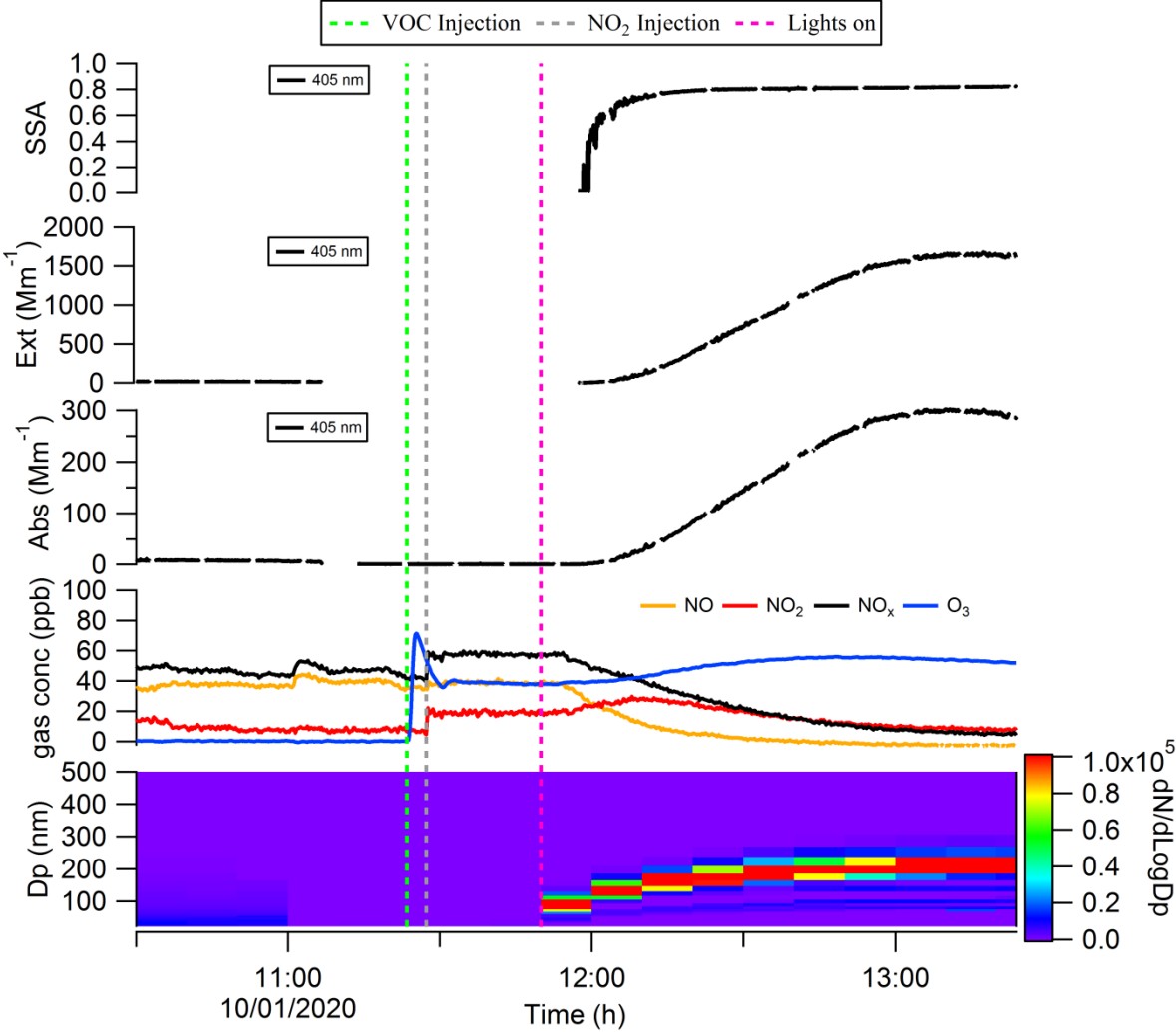

**Fig. 3.** Formation and evolution of the brown carbon with the precursor of cresol and $NO_x$ in the chamber.

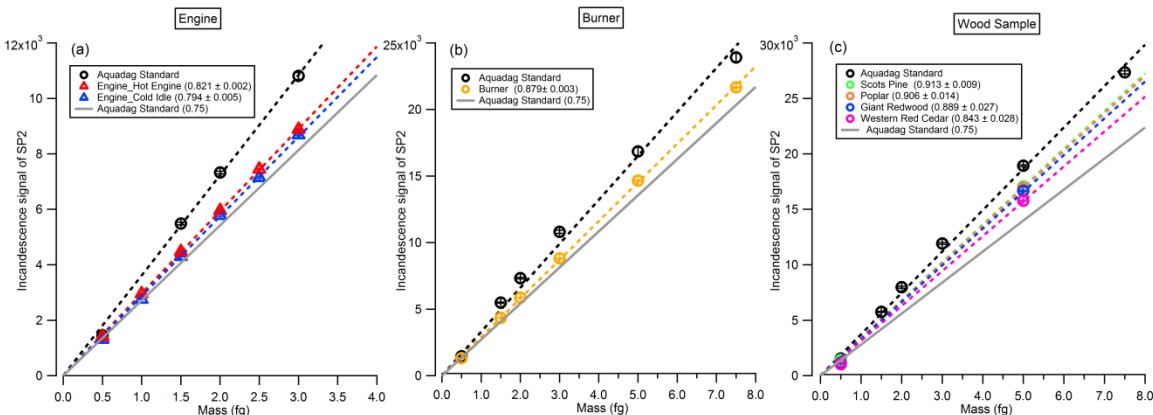

**Fig. 4.** The incandescence signal of the SP2 as a function of the rBC mass, BC particles were selected by the CPMA from: (a) engine emission; (b) burner emission; or (c) wood combustion emission.

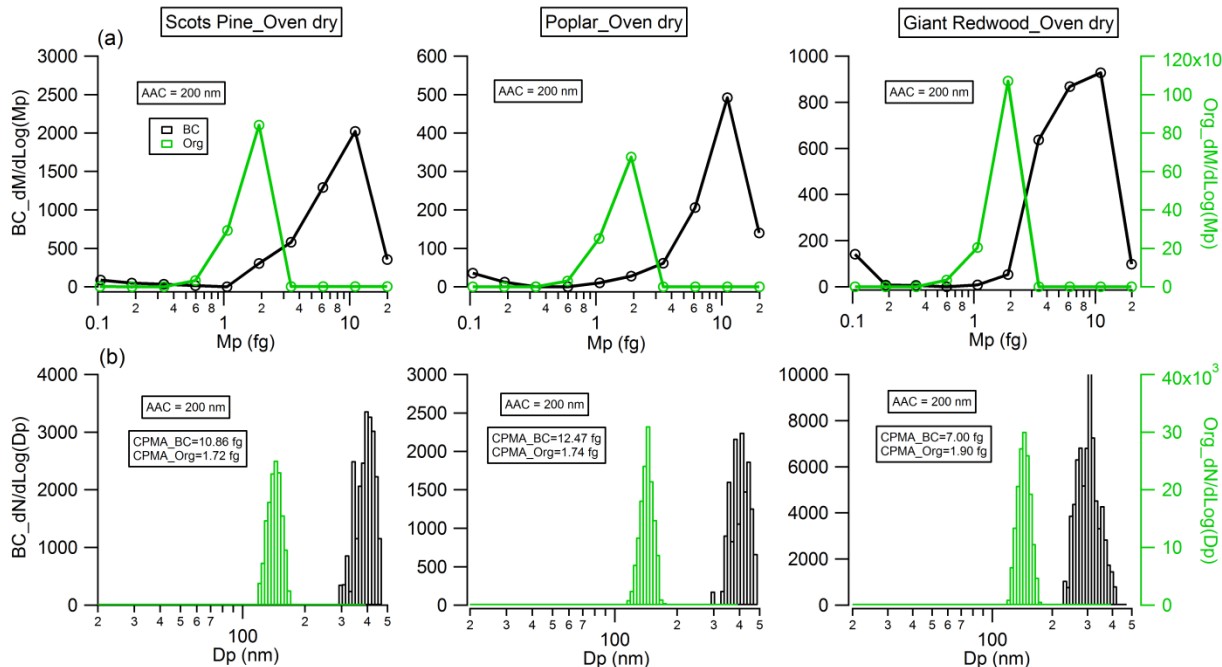

**Fig. 5.** Physical properties of mono-disperse ($D_a$=200 nm) black carbon (flaming phase) and organic particles (smouldering phase) emitted from wood combustion: (a) mass distribution measured by CPMA; (b) particle number distribution measured by the SMPS after the CPMA selected particles at the mass peak of the $D_a$=200 nm particle mass distribution presented in (a).

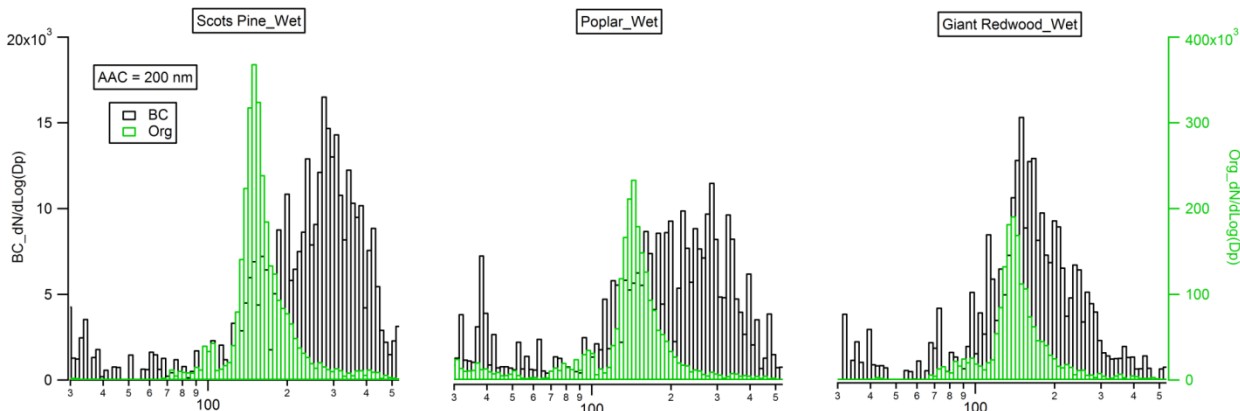

**Fig. 6.** Mobility size distribution of mono-disperse ($D_a$=200 nm) black carbon (flaming phase) and organic particles (smouldering phase) emitted from the combustion of the wet wood samples.

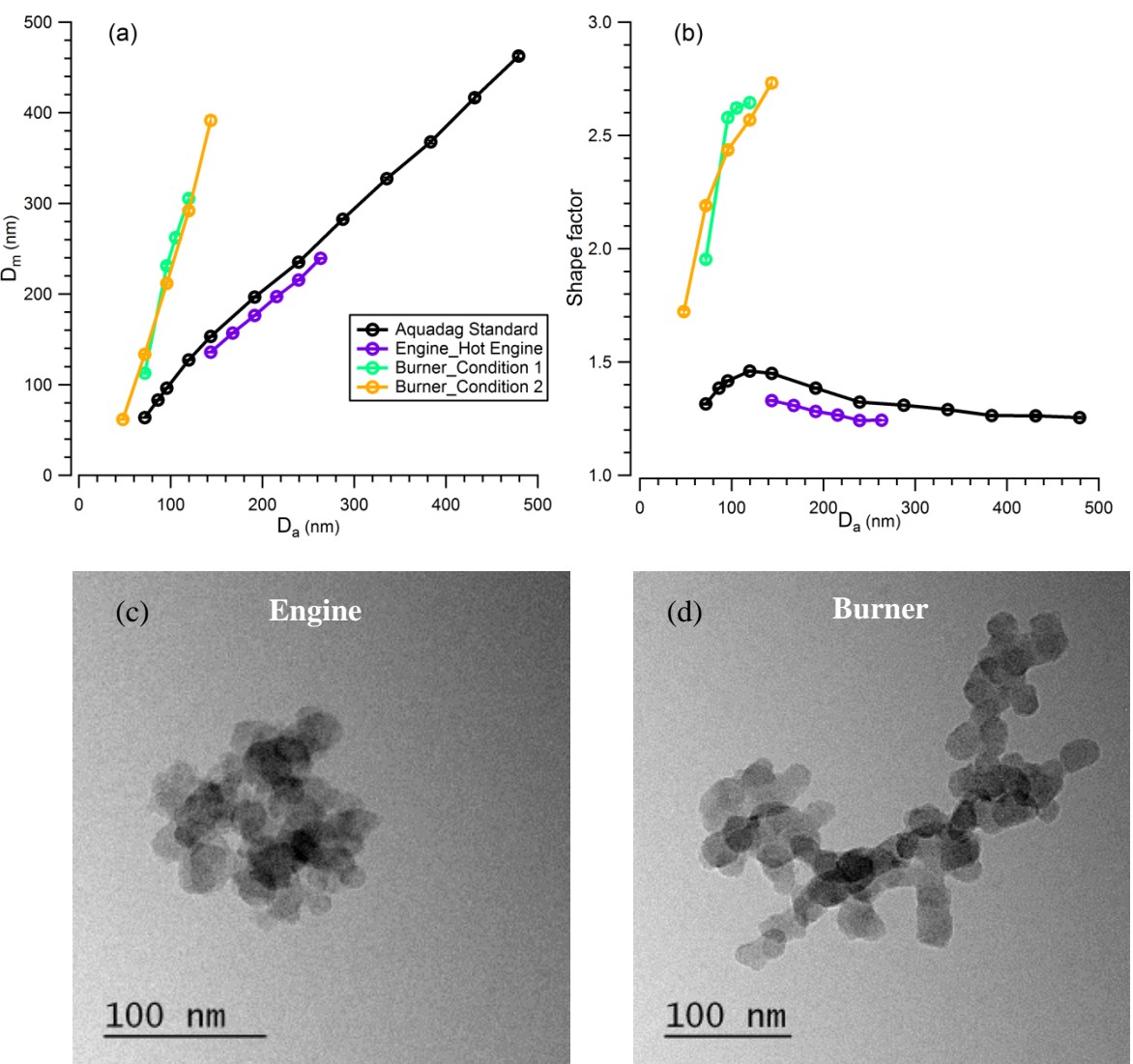

**Fig. 7.** Relationship of: (a) mobility size with aerodynamic size, and (b) shape factor with aerodynamic size, of BC particles from Aquadag standard atomization, diesel engine and flame burner emission; TEM images of BC particles emitted from: (c) diesel engine (hot engine condition) and (d) flame burner (condition 1). The error bar in (a) and (b) is too small to see clearly.

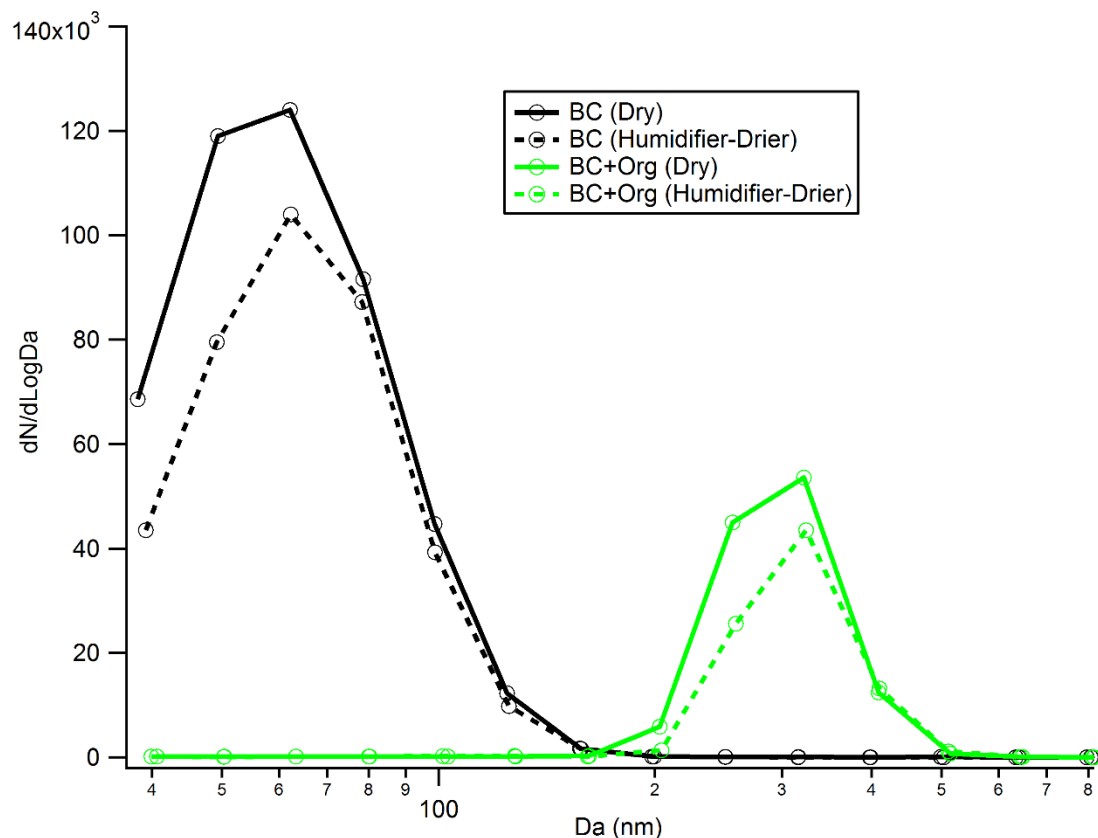

**Fig. 8.** Number size distribution of the bare BC (black line) and organic coated BC particles (green line) before and after they experience 'humidity cycling' (i.e. exposed to 90% RH around 10 s and then dehydrated to 10% RH) process. Da is the aerodynamic diameter.

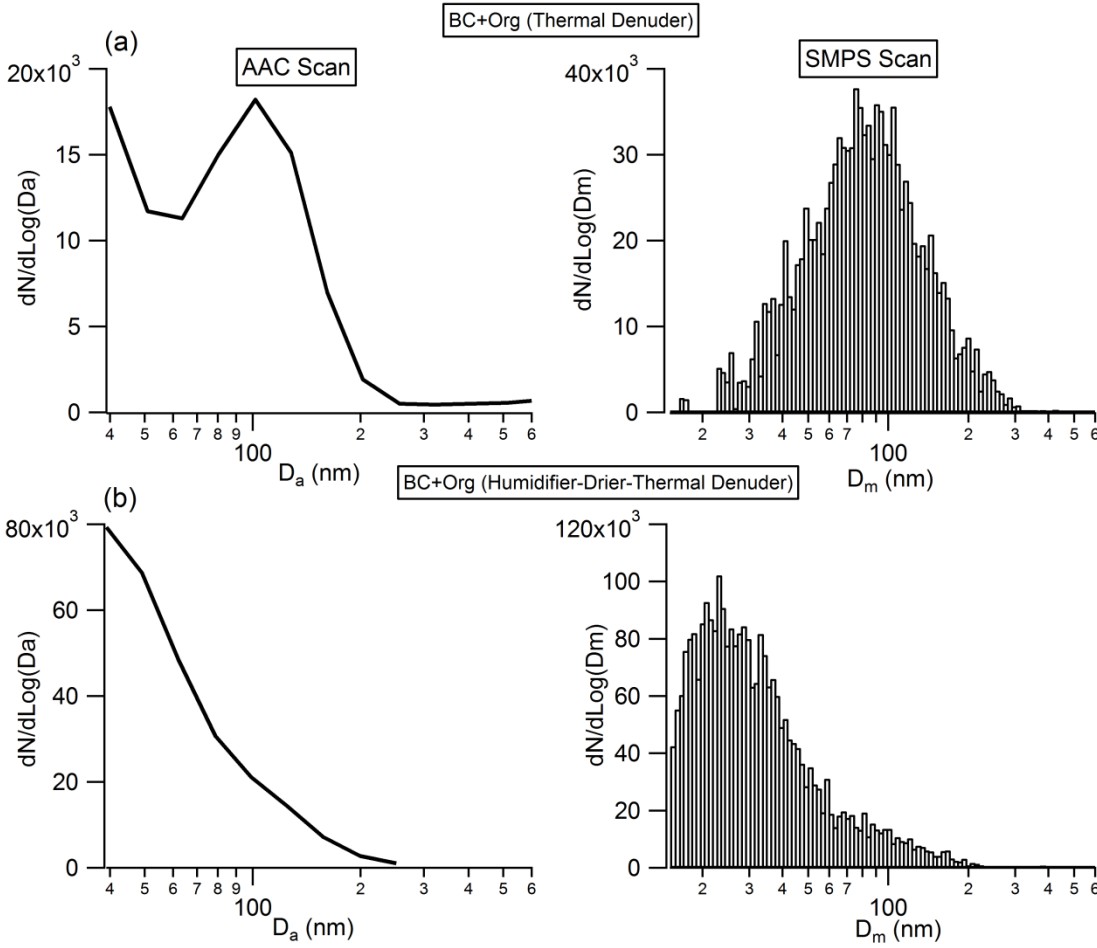

**Fig. 9.** Number size distribution of the: (a) organic coated BC after passing through the thermal denuder to remove coatings; and (b) organic coated BC after experiencing the 'humidity cycling' process and then passing through the thermal denuder to remove coatings.

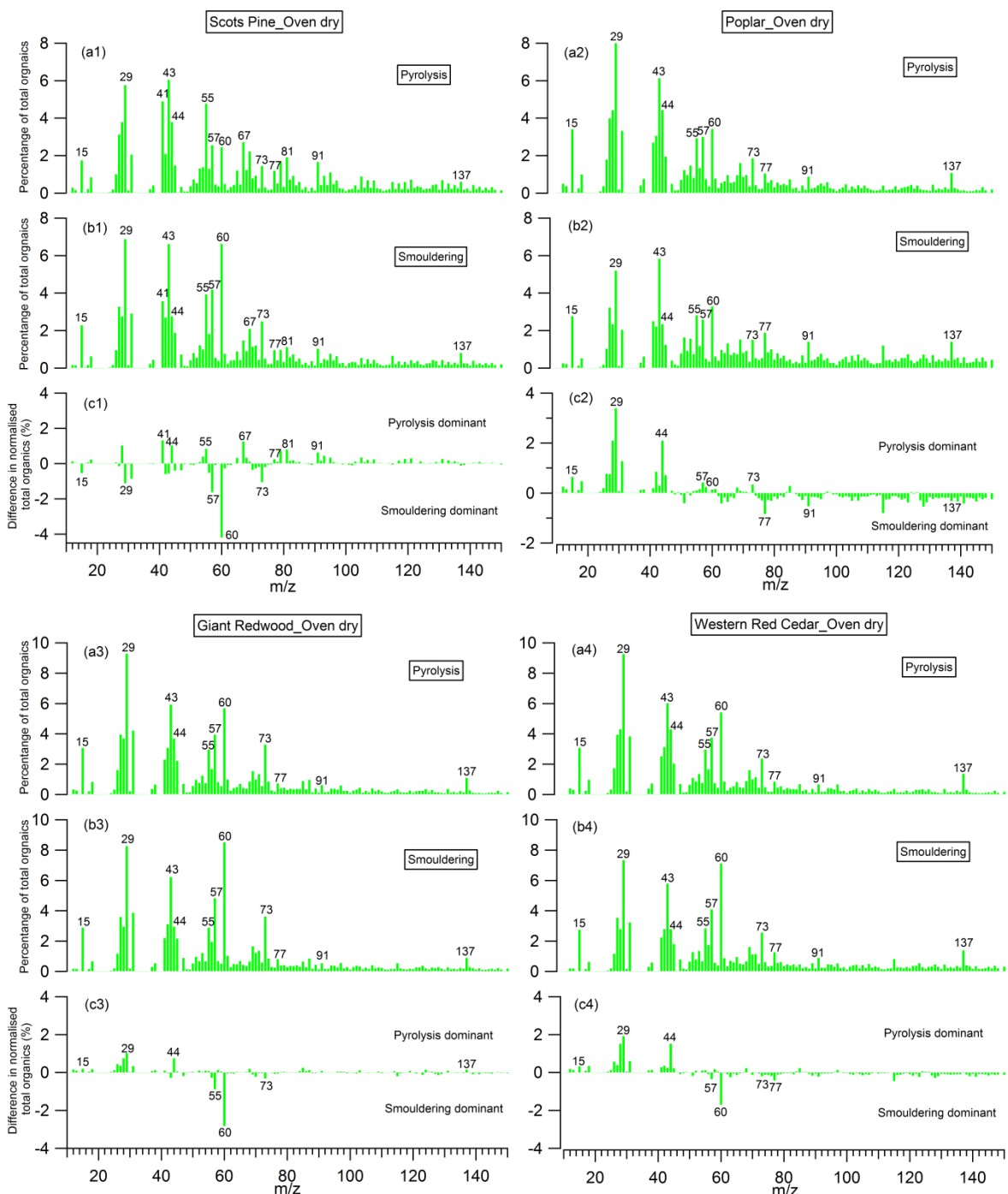

**Fig. 10.** Mass spectra of organic aerosols (OA) presented as a percentage of total OA produced during: (a) pyrolysis phase; (b) smouldering phase; and (c) the difference between them; for the dry wood samples.

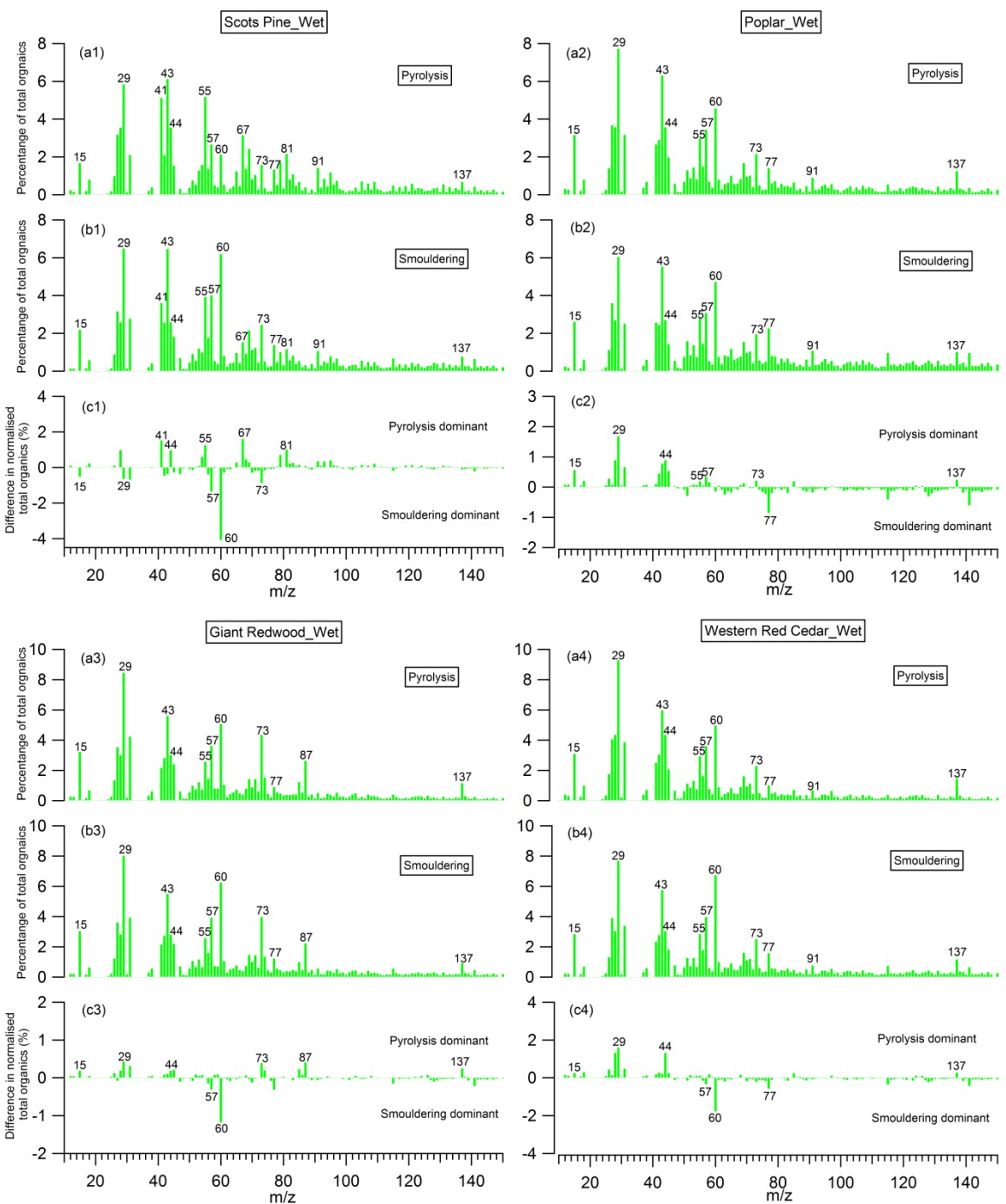

**Fig. 11.** Mass spectra of organic aerosols (OA) presented as a percentage of total OA produced during: (a) pyrolysis phase; (b) smouldering phase; and (c) the difference between them; for the wet wood samples.

# Table and Caption

**Table 1**. Effective density and dynamic shape factor of organic and black carbon aerosols from wood combustion.

| | Flaming phase_BC | | Smouldering phase_Org | |
|---|---|---|---|---|
| | $\chi$ | | $\rho_m$ | |
| | **Oven dry** | **Wet** | **Oven dry** | **Wet** |
| **Scots Pine** | $2.17 \pm 0.04^a$ | $1.85 \pm 0.03^a$ | $1.22 \pm 0.01^b$ | $1.44 \pm 0.02^b$ |
| **Poplar** | $2.10 \pm 0.08^a$ | $1.67 \pm 0.03^a$ | $1.23 \pm 0.01^b$ | $1.52 \pm 0.01^b$ |
| **Giant Redwood** | $1.80 \pm 0.12^a$ | $1.20 \pm 0.02^a$ | $1.32 \pm 0.03^b$ | $1.60 \pm 0.01^b$ |

[a] The BC density of 1.8 g/cm$^3$ was applied.
[b] Assuming organic aerosol is spherical with the shape factor of 1.00.