# Peer review of "Physical and chemical properties of black carbon and organic matter from different sources using aerodynamic aerosol classification"

_Atmospheric Chemistry and Physics, 2021_

## Referee Comment (RC2)

"Physical and chemical properties of black carbon and organic matter from different sources using aerodynamic aerosol classification" by Dawei Hu et al. describes a comprehensive set of measurements from a well-designed and rather clever set of experiments. I believe the measurements are important for the community (especially the BC morphology results) and I look forward to the subsequent paper on optical properties. However, the manuscript is in need of some attention to detail, and in some cases, significant revisions and potential re-analysis of data. I believe this manuscript will be suitable for publication in ACP after these questions and comments are addressed.

**Major Questions and Comments**

Given the scope of the manuscript, the title does not adequately capture the science and I believe a more focused title will help steer readers towards this work when conducting literature searches. With the current title, I would believe that the authors aerosolized a variety of substances and passed them through an AAC, when in reality, the work is geared towards important combustion aerosols and a variety of instrumentation. Consider revising the title to reflect that.

Lines 29-31 of the abstract say "Here we present insights into the physical and chemical properties of the aerosols, with optical properties being presented in subsequent publications" and lines 131-132 of the introduction say "The characterisation and parameterisation of the optical properties of the particles will be the subject of a future publication" yet the introduction seems almost entirely focused on refractive index retrievals. It isn't until line 108 that the authors begin to make the case for morphology and mixing state as a prerequisite for understanding RI. If the optical properties are the subject of a future publication, then the bulk of this introduction belongs in whatever forthcoming paper the authors are preparing, not here. I suggest re-writing the introduction to focus on the literature surrounding the physical and chemical properties that form the basis for the results presented in this manuscript.

The authors claim that their experimental setup enables them to derive refractive index with "much lower uncertainty than has been achieved previously." How is this quantified? How does this particular experimental setup achieve this? What is the current state-of-the-art in RI uncertainty? Finally, how can any statement of this nature be made when this manuscript does not report measurement uncertainties?

The SP2 has been thoroughly described in the literature. Is all of section 2.1.3 necessary? At this stage in the maturation of the SP2, I believe it is more constructive to discuss the artifacts and other measurement caveats of the SP2, for example, Sedlacek et al. (2018). How were these charring artifacts handled in data analysis analysis?

The SMPS has an even longer history of prior publication. Section 2.1.6 can be significantly shortened to include only the operational parameters such as scan time and flow rates, and the resulting bounds of the size distribution you are able to measure. Once this has been described, the manuscript would benefit from a discussion on the uncertainties introduced by the SMPS. Finally,

please describe the inversion algorithm used. Is it the typical inversion algorithm that comes with the SMPS software, or is it custom? If it is custom, it should be discussed and cited.

Why was 180 °C chosen for the thermal denuder? The optimal internal temperature for a thermal denuder is the subject of ongoing research, yet the consensus is that it must be chosen based on some optimization of experimental parameters. In this case, I would guess that this optimization corresponds to complete volatilization of coating materials so that BC alone can be studied. However, it is never explicitly stated what the reasons are for choosing 180 °C, nor is the method by which the authors arrived at this temperature. For example, Sumlin et al. (2018) describes an approach using four different TD temperatures in an attempt to volatilize different mass fractions of organic matter. Please discuss the approach to choosing 180 °C.

Furthermore on the subject of the TD, the authors state that the inner diameter of the heating zone is 0.15 meters. This is rather large, and it is almost certain that there is a temperature gradient across the radius. The temperature along the central axis may be several degrees colder than at the walls, and aerosol travelling at different points along the radius will experience varying degrees of thermal processing. Was this temperature gradient measured? How does this gradient affect the analysis?

Finally, on the subject of the TD, the authors state that the residence time in the heating section was approximately 31 seconds. Given the measurements provided for both the inlet and outlet (0.037 m ID tube) and the heating section (0.15 m ID) it seems that the authors did not account for fluid velocity change within the heating section. If one considers only 1 liter min-1 flow through the 0.037 m ID tube, across the 0.51 m heating section (assuming steady and conservative flow), one arrives at 32.9 s, which is "approximately" 31 s as stated in the manuscript. However, that assumes that fluid velocity is constant throughout the TD – it is not, since the ID in the heating section is different than the inlet and outlet. The mean residence time in the heating section is likely somewhere on the order of nine minutes. Given this, and my previous comment, I am unsure that conclusions from the TD experiment are represented accurately. It is likely that this data can be re-processed, or at least re-interpreted, but the data may have been handled according to an incomplete understanding of the experimental setup. This must be corrected or addressed in the manuscript.

It seems that all measurements are reported without uncertainties. The authors should report uncertainties or justify their exclusion.

Finally, please make sure all references are properly formatted and include DOI numbers, if possible.

**Minor questions and comments**

The name "Manchester Aerosol Chamber" is unfortunate, since it shares an acronym (MAC) with a commonly studied aerosol optical parameter, the mass absorption cross-section. Unless this acronym has been used for your chamber multiple times in other publications, I would suggest you consider renaming it.

Several parts of the Experimental Methods section contain results, especially section 2.3.4. I suggest re-writing these sections to focus on the methods, and thoroughly discuss the results in section 3.

The headings for sections 2.1.1, 2.1.2, 2.1.3, 2.1.5 and 2.1.6 should be spelled out, for consistency with other 2.1.X section headings.

The description about the AAC in section 2.2.1 is overly subjective. Furthermore, the AAC only produces a monodisperse aerosol in the sense of aerodynamic diameter. This is not "truly" monodisperse, since there are other measures of aerosol diameter that run contrary to this qualifier.

Line 278: The authors may consider stating the material that the space blanket (mylar, I'm guessing) is made from so that readers unfamiliar with the term will know what was used.

Line 281: How was the illumination evaluated?

Line 357: Can the authors comment on what effect resin content might have on the resulting aerosols?

Line 366-367: How was the "sufficient concentration" to ignite the pyrolysate determined?

Line 380: Change "exemplar" to "example".

Line 404: Change "300 s of the ignition" to "300 s after the ignition."

Line 430: I suggest specifying the NO2 came from a "compressed gas cylinder". Also, what was the concentration of NO2, and what was the balance gas?

Line 432: Change "desirable" to "desired".

Lines 451-454: I suspect that the increase in ozone concentration is attributed to the absorption of UV photons by  $O_2$  in the chamber – this is a common method to produce  $O_3$  from  $O_2$ . I cannot find any references in the literature on  $O_3$  production due to UV absorption by cresol. If the authors can provide a reference for  $O_3$  production by cresol photochemistry, please do so.

Line 464: Change "desirable" to "desired". What was the desired concentration?

Line 470: "particulate" is an adjective. Change to "particles".

Line 471-472: What size did the particles stabilize to?

The caption on Figure 1 refers readers to the main text to thoroughly understand what is meant by the different groupings of instruments, however, the first reference to the figure in section 2 does not clarify the groupings whatsoever. The figure is therefore unclear: were different groupings of instruments used during different experiments? Were they grouped in this way during data analysis to calculate specific parameters? This needs to be clarified both in the figure and its caption, and in the main text.

**References**

Sedlacek et al., Formation of refractory black carbon by SP2-induced charring of organic aerosol, Aerosol Sci. & Tech, 52:12, 1345-1350, 10.1080/02786826.2018.1531107, 2018.

Sumlin et al., Density and Homogeneous Internal Composition of Primary Brown Carbon Aerosol, Environ. Sci. Technol. 2018, 52, 7, 3982-3989, 10.1021/acs.est.8b00093, 2018.

---

## Author Comment (AC1)

**Response to comments of anonymous referees  # 1**

**Review of the manuscript titled: "Physical and chemical properties of black carbon and organic matter from different sources using aerodynamic aerosol classification" by Dawei Hu et al.**

The paper describes laboratory measurements of the physical properties of black carbon particles emitted from different sources. The paper is overall well written (with some relatively minor issues as discussed later), and the approach seems quite comprehensive and, for the most part, sound. The results provided by this study are important for the community and I would like to see them published. The paper requires some significant but relatively straightforward revisions, after which, the paper can be most probably published.

We thank the reviewer for the positive comments on our manuscript. In the revised version, we have addressed the comments listed below.

**General comments**

- A good part of the introduction focuses on optical properties and refractive indices, but then at the end of it, the authors mention that the optical properties are not the subject of the current paper. I would suggest refocusing the introduction on the topic of the paper.

  The introduction is rephrased in the revised manuscript (line 63-161 on page 3-7). More discussions regarding the physical properties of particles are added in line 80-108 on page 4 and 5.

  "Although BC and BrC are very important for climate, they are poorly represented in atmospheric models (Zuidema et al., 2016). This is in part due to the complex microphysical properties of BC and the lack of accurate refractive index (RI) descriptions for both BC and BrC (Liu et al., 2020). Fresh soot particles often exist in the form of aggregates composed of primary spherules with an irregular and highly fractal geometry (Xiong and Friedlander, 2001;Wentzel et al., 2003). The morphology of these aggregates change markedly during the atmospheric aging process, influencing the corresponding particle size and optical properties (Zeng et al., 2019;Zhang et al., 2008). For example, after condensation of gaseous species such as

sulfuric acid or water (under high relative humidity (RH) environments) on soot particles, or coagulation with the pre-existing particles, soot particles can experience restructuring and the shape of the soot particles becomes more similar to a spherical particle (Zhang et al., 2008). The morphology of BC particles can be measured directly by using Scanning Electron Microscopy (SEM) or Transmission Electron Microscopy (TEM) (Fu et al., 2006;Chen et al., 2018;Ellis et al., 2016). However, the SEM/TEM approach only provides particle shape information in two dimensions and do not provide real time characterisation. Alternatively, the particle morphology can be determined by measuring its size and mass with different techniques (Chen et al., 2018;DeCarlo et al., 2004). A conventional approach is to classify particles (generally using a differential mobility analyser, DMA, to select monodisperse particles on their mobility size) and then measure particle mass using a particle mass analyser (Zhang et al., 2008;Park et al., 2003;Park et al., 2004a;Park et al., 2004b;Chen et al., 2018) (Wu et al., 2019). From the resulting information about particle mass for different particle mobility sizes, the dynamic shape factor ($\chi$, defined as the ratio of the drag force on the particle divided by the drag force on the particle's volume equivalent sphere) and fractal dimensions ($D_f$) can be retrieved (DeCarlo et al., 2004). Mobility-mass fractal dimension ($D_{fm}$) has been reported over a wide range of 2.2-2.8 for diesel exhaust particles (Park et al., 2004b). $D_{fm}$ has been reported as higher than the $D_f$, - defined as the scaling exponents between the radius of gyration of an aggregate and the radius of primary spherules composing the aggregate - but the two are not always directly equivalent, particularly in the transition regime (Sorensen, 2011)."

- The excessive use of emphatic words in the abstract/introductions such as "pioneering", "authoritative", "novel", etc. detracts from the undoubted value of the work. I would suggest removing these terms that are just irritating and add nothing to the paper.

  These terms are removed in the revised manuscript.

- How do multiple charges affect the mass measurements provided by the CPMA and how is that accounted for?

  In this study, as the particles from each source (either engine, wood combustion or flame burner) will have a single effective density for a given size, the AAC will

deliver particles of only a single physical size. Therefore there will be no larger particles available to be multiply charged prior to sizing by the CMPA.

- Results are reported without uncertainties, making comparisons, and the understanding of the significance of the results difficult. Please estimate potential uncertainty bounds (both statical as well as biases) for all the quantities reported or calculated including chemical, morphological, or other physical quantities such as densities, fit slopes, shape factors, etc.

The uncertainties are calculated and added for the data regarding the shape factor of BC (Table 1 and Figure 7, The error bar in Fig. 7(a) and Fig. 7(b) is too small to see clearly), material density of organics (Table 1), fit slopes for the SP2 incandescence signal calibration (Figure 4, the uncertainties refer to fitting precision based on the standard error in the regressed slope). The noise to signal ratio ($<0.005$) is also provided of the organic mass spectrum. In addition, all these above uncertainties are now quoted in the main text (e.g., in Sect. 3.4 for the reported dynamic shape factors and densities, line 710 for the noise to signal ratio, and lines 765-766 for the SP2 incandescence signal correction factor).

- Some grammatical and tense consistency checks would be advisable (limited examples in the specific comments next).

The grammar and tense have been checked for consistency throughout the manuscript.

- References are somewhat scarce and myopic, neglecting some important related work especially on the AAC/CPMA/DMA use, BC morphology, and SP2 signal interpretation. I did not provide too many specific examples below just because there is a lot of work out there that seems very relevant to this study.

More related references (listed below) have been added and discussed in the introduction of the revised manuscript.

[revised manuscript text omitted]

**Specific comments**

Line 44: "This implies" or "This suggests", how certain are the authors about the following statement?

"This implies" is modified to "This suggests"

Line 51: Please provide uncertainty bounds for these values, otherwise it is hard to understand if the later statement (on line 55) on the difference from the 0.75 value might be justified; in other words, is the difference significant?

The uncertainty is analysed and discussed in the revised manuscript.

The sentence is modified to: "A correction factor is defined as the ratio of the incandescence signal from an alternative BC source to that from the Aquadag standard, and took values of 0.821 ± 0.002 (or 0.794 ± 0.005), 0.879 ± 0.003 and 0.843 ± 0.028 to 0.913 ± 0.009 for the BC particles emitted from the diesel engine running under hot (or cold idle) conditions, the flame burner and wood combustion, respectively."

Line 50-54 on page 2 and 3.

Line 75: The statement that the absorption coefficient for BC is wavelength-independent is incorrect, the typical dependence, as extensively reported in the literature, is often expressed as a power law with an exponent of about -1 (which is still a strong wavelength dependence, although weaker than that of brown carbon). What is often assumed (but probably also not always true) is that the imaginary part of the index of refraction is wavelength-independent (or at least not very strongly dependent). BrC also has an absorption that is wavelength dependent just with an exponent that is significantly larger, in absolute value, than that of BC.

The sentence is modified to: "Typically, the light absorption coefficient for BC and BrC is wavelength dependent over the visible spectrum, with BrC exhibiting a stronger wavelength dependence characterised by increasing absorption at progressively shorter visible wavelengths (Kirchstetter et al., 2004;Corbin et al., 2019;Voliotis et al., 2017)."

Line 76-79 on Page 3 and 4.

Line 95: This is an interesting approach but it is hardly pioneering, I would call this incremental in a very positive sense (see, for example, the work by the Olfert's group, or others). I suggest removing this exaggerated adjective and point to existing literature. Same in line 99.

The term "pioneering" is removed.

The sentence is revised to: "To address the issues mentioned above, the Soot Aerodynamic Size Selection for Optical properties (SASSO) project utilised the Aerodynamic Aerosol Classifier (AAC) to classify particles according to aerodynamic diameter for size and mass distribution measurements and optical evaluation (Tavakoli et al., 2014). Specifically, SASSO has used the AAC size selection of emissions from wood burning, diesel combustion and secondary organic aerosol (SOA) formation, prior to optical measurements using cavity ring-down and photoacoustic spectroscopy with the EXtinction, SCattering and Absorption of Light for AirBorne Aerosol Research (EXSCALABAR) instrumentation, custom-built by the Met Office (Cotterell et al., 2020;Cotterell et al., 2019;Davies et al., 2018)."

Line 133-141 on page 6.

Line 97: How do the authors determine themselves that the method is "authoritative"? That, if true, should be a judgment left to the community.

The word "authoritative" is deleted.

Line 111: In what way does the SP2 provide information about the morphology? The information is likely limited and subject to large uncertainties. Several papers have been published on the topic, some in contrast with others.

The SP2 may provide some morphology by analysing the relative peak position between scattering and incandescence signal, however this technique is not quantitative which is subject to considerable uncertainties, and literatures reported contrasting results. We therefore use size/mass or TEM measurements to obtain the morphology rather than using the SP2 itself.

The "morphology" term is removed and the sentence is revised to:

"Important additional considerations in the retrieval of refractive indices from optical spectroscopy data are the aerosol morphology (described above) and mixing state. The mixing state can be probed using the Single Particle Soot Photometer (SP2), which can measure the refractory BC (rBC) mass content and optical size of individual particles. However the SP2 needs an empirical calibration to retrieve the rBC mass from the

incandescence signal (Laborde et al., 2012a). The conventional method to calibrate the incandescence channel of SP2 is using size selected Aquadag standards (Acheson Inc. USA) and then correcting to a calibration representative of ambient rBC by a constant factor of 0.75 (Laborde et al., 2012b). However, few experiments since have independently verified this across various soot types."

Line 123-132 on page 5 and 6.

Line 116: Consider rewording "which makes the complexity of the calibration methods" to "which makes the calibration methods complex" or "challenging"

This is revised.

Line 118: Change "corrected" to "correcting"

This is revised.

Line 135 "to to" -> "to"

This is revised.

Line 146: The AAC select aerosol by aerodynamic size; so, aerosol particles passing through it are indeed monodisperse in terms of aerodynamic size, but that does not mean that the output distribution is mondisperse in every size measure; for example, particles of the same mass (and therefore mass-equivalent diameter) could have very different aerodynamic size depending on their morphology. So, the term monodisperse here is ambiguous. And it all depends on the property one wants to measure (for example, absorption mostly depends on mass).

This sentence is modified to: "The AAC (Cambustion Ltd, Cambridge, UK) is used to select aerosols within a narrow range of aerodynamic diameters and does not suffer from the issue of multiple charges that affects selection using instruments such as the CPMA and DMA."

Line 180-182 on page 8.

We modified the term of "monodisperse particles" associated with AAC to "particles classified within a narrow range of aerodynamic diameter" throughout the whole manuscript.

Line 200: How well does an optical size measurement calibrated with PSLs perform on fractal-like black carbon particles? Is the size an optical equivalent to a spherical PSL particle? That should be mentioned as the meaning of "size" for a fractal-like particle is always quite ambiguous (see the previous comment as well).

A sentence is added for further clarification:

"The particle size can be determined by detecting the laser signal scattered by particles, with the scattering intensity maximum related to the optical particle diameter through a calibration using polystyrene latex spheres. The optical size of BC-containing particles is determined by matching the measured scattering signal with calculations from light scattering calculations assuming a core-shell structure (core-shell Mie theory) (Moteki and Kondo, 2007)."

Line 229-234 on page 10.

Line 217: Change verb in the sentence "The instrument operation and data analysis of HR-AMS has been…" to "The instrument operation and data analysis of HR-AMS have been…" for number consistency.

This is revised.

Section 2.1.5: The CPMA, using an electric field, also suffers from the issue of multiple charges as in the case of the DMA, this should be mentioned. Also, what charge neutralizer was used for the CPMA should be mentioned for consistency with the following description of the SMPS.

The following sentence is added for further clarification.

"As the CPMA uses an electrical classification method to select particles, it also suffers from an issue of multiple charging similar to a DMA, and it has problems associated with a fraction of the uncharged particles that are also transmitted, particularly at the lower rotation speeds. In this study, an electrical ioniser (MSP Corp., USA) was used for wood combustion experiments and a $^{90}$Sr radioactive ioniser was used for chamber experiments to neutralize particles before they were sampled by the CPMA."

Line 269-274 on page 12.

Line 259: It would be good to provide a reason behind the choice of the denuder temperature set point.

For the purpose of this study, we wanted to remove as much of the organic coating material from the combustion-generated particles as possible and 180 °C is the maximum temperature for our self-built thermal denuder.

The following sentence is added for further clarification.

"In this study, the purpose of the TD is to remove as much of the organic coatings from the combustion-generated particles as possible, rather than probing the volatility properties of the coating. Therefore, all heating zones of the TD were set to their maximum temperature of 180 °C. This upper temperature is lower than that achieved by other commercial TD units and minimises the risks of charring."

Line 304-309 on page 13.

Line 285: Suggest changing "can be" to "to be"

This is revised.

Line 285-286: Do the authors have a more quantitative measure of the aerosol loss rate?

Yes, the wall losses of particles inside the Manchester aerosol chamber was investigated and the corresponding results is under review in AMT (Shao et al., 2021). Briefly, a series of experiments were conducted to investigate the size-resolved particle lifetimes under various humidity and mixing conditions using ammonium sulfate particles, which was introduced to the chamber and left in the dark at the desired RH and temperature conditions for ≥4 hours. The mean number and mass wall loss rates were estimated as $9.17 \pm 1.3$ and $8.16 \pm 1.5 \times 10^{-5}$ $s^{-1}$, respectively. More details can be found in section 3.5 in Shao et al. (2021).

The following sentence is added in the revised manuscript.

"The relatively large volume of the chamber allows the dilute sample to be held for several hours without significant aerosol removal from wall losses, allowing the study of particles introduced directly or formed within the chamber over a period of several hours. The mean

number and mass wall loss rates of particles inside the chamber were estimated as $9.17 \pm 1.3$ and $8.16 \pm 1.5 \times 10^{-5}$ s$^{-1}$, respectively (Shao et al., 2021)."

Line 335-339 on page 15.

"There are two possibilities: (1) The BC cores retained their structure throughout humidity cycling process; or (2) As shown in Figure 8, the coatings on BC particles are very thick and therefore dominate the particle, rendering our size measurement approach insensitive to any changes in BC core size from restructuring. The size of the bare BC particles is around 68 nm, but after coating by SOA, the size of the coated BC particles reaches up to around 300 nm. Due to the very large coating thicknesses of the coated BC particles, even if the BC cores were restructured during the humidity cycling process, any changes were not reflected in the overall particle size as this was dominated by the contribution from the coating organics."

Line 490: This is a very small diameter. How large were the monomers in these BC particles, and how many monomers typically in an aggregate? Were these particles made of only a very few monomers?

As shown the TEM images in Fig. 7(c), the primary spherule size of the BC particles from the diesel engine is around 10-15 nm. However, we do not have micrographs for these specific experiments, so cannot report on the precise numbers of monomers in this case.

Line 501: Maybe replace "improve" with "improving"?

This is revised.

Figure 6: Especially for Aquadag (but it might be slightly visible also in some of the other BC types), there seems to be a slight negative curvature in the graphs (especially visible in the center and right graphs). What is the reason for such a change in slope? One could study these changes of the slope by graphing residuals plots. I believe Aquadag comes already compacted; is it possible that the compacted morphology "shields" the aggregate lowering the incandescence signal at higher masses with respect to what might be expected for not compacted BC particles of the same mass, resulting in the negative curvature?

We believe the deviation from linear to only be very slight and small compared to the variations in slopes, which is the key result being presented here. While the effect described by the reviewer may be plausible, we should point out that there may be a competing effect if resonances occur within spherical particles. But whichever way, it is generally assumed that the interactions between the BC and the incident light occur within the Rayleigh regime (Moteki and Kondo, 2007), where either effect should be very minor. Regardless, we do not believe that we can conclude anything firm based on this data, so consider this outside of the paper's scope.

Reference:

Moteki, N., and Kondo, Y., 2007. Effects of Mixing State on Black Carbon Measurements by Laser-Induced Incandescence. Aerosol Science and Technology. 41, 398-417, http://doi.org/10.1080/02786820701199728.

Lines 510 to 526: These are very interesting results, but uncertainty bounds should be reported to understand how significant these differences are. How the uncertainties (both statistical and systematic) are estimated, should also be carefully described.

The uncertainties are calculated and added and discussed in the revised manuscript.

"In this study, the incandescence signal of the SP2 was measured for BC particles from catalytically stripped diesel engine exhaust emissions, an inverted flame burner, and controlled flaming wood combustion, respectively, and compared with that measured from an Aquadag standard. The uncertainties here refer to precision of the fitted parameters reported by the Igor Pro fitting algorithm, based on analysis of residual data. As shown in Fig. 4, for the BC particles emitted from the diesel engine under hot engine and cold idle conditions (Fig. 4(a)), the slopes of the incandescence signal with BC mass are 0.821 ± 0.002 and 0.794 ± 0.005 times of that measured from the Aquadag standard, respectively. Note that while some deviation from a perfect linear response is noted, this is small compared to the variation in slopes, so represents a minor source of uncertainty in comparison. These correction factors are 9.4% and 5.6% different with the common value of 0.75 (with the uncertainty less than 5%) recommended by Laborde et al. (2012b) when deriving the mass concentration of BC emitted from diesel engines. For the BC particles generated from the flame burner (Fig. 4(b)), the correction factor is 0.879 ± 0.003. Meanwhile, for the BC particles emitted from the

flaming phase during the combustion of Scots pine, Poplar, Giant Redwood or Western red cedar, the correction factors are 0.913 ± 0.009, 0.906 ± 0.014, 0.889 ± 0.027 or 0.843 ± 0.028, respectively. We stress that, for the SP2 calibrations here from wood combustion emissions, the BC particles were not treated with a catalytic stripper before sampling by the SP2. While coating materials may char under 1064 nm to produce refractory black carbon and therefore cause overestimates in the incandescence signal (Sedlacek et al., 2018), as shown in Fig. 2(c), the BC particles generated at the beginning of the flaming phase contained almost no organic species, with $r_{OA}$ values less than 0.05. Even if this OC were to be converted to EC with 100% efficiency (which we consider to be highly unlikely), this would represent a very small error.

The differences in the correction factors derived in this study with the default value of 0.75 are 9.4% (5.6%), 17.2% and 12.4-21.7% for the BC particles emitted from engine with hot engine (or cold idle) condition, flame burner and wood combustion, respectively. We recommend that future studies utilizing the SP2 for rBC mass concentration measurements use SP2 calibrations with the same type of BC as that to be studied."

Line 554-581 on page 24 and 25.

Lines 516-521: This means that some organics still coat the BC particles, even if in a small amount, correct? Is it possible that some of this organic would char and generate an incandescence signal like that of BC? See, for example, Sedlacek, et al. Atmos. Chem. Phys. 18: 11289-11301 (2018).

The reviewer is correct in saying that the BC particles produced from the flaming wood combustion may have some organic components, particularly given that the combustion-generated particles were not treated with a catalytic stripper before sampling by the SP2. Based on the study by Sedlacek et al. (2018), some of coating materials may char under 1064 nm to produce the refractory black carbon and then overestimate the incandescence signal. However due to experimental limitations, we were unable to confirm the presence of this phenomenon. In this study, as the measured organic mass fraction in the BC particles is less than 0.05, the charring effect would be very limited.

A sentence is added to discuss the charring effect in line 570-576 on page 25.

"We stress that, for the SP2 calibrations here from wood combustion emissions, the BC particles were not treated with a catalytic stripper before sampling by the SP2. While coating

materials may char under 1064 nm to produce refractory black carbon and therefore cause overestimates in the incandescence signal (Sedlacek et al., 2018), as shown in Fig. 2(c), the BC particles generated at the beginning of the flaming phase contained almost no organic species, with $r_{OA}$ values less than 0.05. Even if this OC were to be converted to EC with 100% efficiency (which we consider to be highly unlikely), this would represent a very small error."

Section 3.2: As mentioned in the general comments, here (as in other places in the paper) a comparison is difficult without having a good estimate of how certain these reported values might be.

The uncertainties are calculated and added in Table 1, Figure 7 and the manuscript.

Lines 644-645: What does it means that "the peaks are most dominated in the smouldering phase"? Do they mean "are most dominant in the smouldering phase" or something else? Also, check tense consistency with just a couple of lines earlier

Yes, it should be "the peaks are most dominant in the smouldering phase". This is modified in the revised manuscript.

The tense consistency is checked in the revised manuscript.

Line 653: "in" in front or "contrast".

This is revised.

Lines 669 – 671: "clear difference… was" or "clear differences … were" but not "clear difference… were"

This is revised.

---

## Author Comment (AC2)

**Response to comments of anonymous referees #2**

"Physical and chemical properties of black carbon and organic matter from different sources using aerodynamic aerosol classification" by Dawei Hu et al. describes a comprehensive set of measurements from a well-designed and rather clever set of experiments. I believe the measurements are important for the community (especially the BC morphology results) and I look forward to the subsequent paper on optical properties.

**We thank the reviewer for the positive comments on our manuscript**

However, the manuscript is in need of some attention to detail, and in some cases, significant revisions and potential re-analysis of data. I believe this manuscript will be suitable for publication in ACP after these questions and comments are addressed.

In the revised version, we have addressed the comments listed below.

**Major Questions and Comments**

Given the scope of the manuscript, the title does not adequately capture the science and I believe a more focused title will help steer readers towards this work when conducting literature searches. With the current title, I would believe that the authors aerosolized a variety of substances and passed them through an AAC, when in reality, the work is geared towards important combustion aerosols and a variety of instrumentation. Consider revising the title to reflect that.

**We thank the reviewer to point this out.**

The title is modified to: "Physical and chemical properties of black carbon and organic matter from different combustion and photochemical sources using aerodynamic aerosol classification"

Lines 29-31 of the abstract say "Here we present insights into the physical and chemical properties of the aerosols, with optical properties being presented in subsequent publications" and lines 131-132 of the introduction say "The characterisation and parameterisation of the optical properties of the particles will be the subject of a future publication" yet the introduction seems almost entirely focused on refractive index retrievals. It isn't until line 108 that the authors begin to make the case for morphology and mixing state as a prerequisite

for understanding RI. If the optical properties are the subject of a future publication, then the bulk of this introduction belongs in whatever forthcoming paper the authors are preparing, not here. I suggest re-writing the introduction to focus on the literature surrounding the physical and chemical properties that form the basis for the results presented in this manuscript.

We thank the reviewer to point this out.

The introduction is rephrased in the revised manuscript (line 63-161 on page 3-7). More discussions regarding the physical properties of particles are added in line 80-108 on page 4 and 5.

"Although BC and BrC are very important for climate, they are poorly represented in atmospheric models (Zuidema et al., 2016). This is in part due to the complex microphysical properties of BC and the lack of accurate refractive index (RI) descriptions for both BC and BrC (Liu et al., 2020). Fresh soot particles often exist in the form of aggregates composed of primary spherules with an irregular and highly fractal geometry (Xiong and Friedlander, 2001;Wentzel et al., 2003). The morphology of these aggregates change markedly during the atmospheric aging process, influencing the corresponding particle size and optical properties (Zeng et al., 2019; Zhang et al., 2008). For example, after condensation of gaseous species such as sulfuric acid or water (under high relative humidity (RH) environments) on soot particles, or coagulation with the pre-existing particles, soot particles can experience restructuring and the shape of the soot particles becomes more similar to a spherical particle (Zhang et al., 2008). The morphology of BC particles can be measured directly by using Scanning Electron Microscopy (SEM) or Transmission Electron Microscopy (TEM) (Fu et al., 2006; Chen et al., 2018; Ellis et al., 2016). However, the SEM/TEM approach only provides particle shape information in two dimensions and do not provide real time characterisation. Alternatively, the particle morphology can be determined by measuring its size and mass with different techniques (Chen et al., 2018;DeCarlo et al., 2004). A conventional approach is to classify particles (generally using a differential mobility analyser, DMA, to select monodisperse particles on their mobility size) and then measure particle mass using a particle mass analyser (Zhang et al., 2008; Park et al., 2003; Park et al., 2004a; Park et al., 2004b; Chen et al., 2018) (Wu et al., 2019). From the resulting information about particle mass for different particle mobility sizes, the dynamic shape factor  $(\mathbf{x}, \mathbf{d})$  defined as the ratio of the drag

force on the particle divided by the drag force on the particle's volume equivalent sphere) and fractal dimensions ( $D_f$ ) can be retrieved (DeCarlo et al., 2004). Mobility-mass fractal dimension ( $D_{fm}$ ) has been reported over a wide range of 2.2-2.8 for diesel exhaust particles (Park et al., 2004b).  $D_{fm}$  has been reported as higher than the  $D_{f}$ . - defined as the scaling exponents between the radius of gyration of an aggregate and the radius of primary spherules composing the aggregate - but the two are not always directly equivalent, particularly in the transition regime (Sorensen, 2011)."

The authors claim that their experimental setup enables them to derive refractive index with "much lower uncertainty than has been achieved previously." How is this quantified? How does this particular experimental setup achieve this? What is the current state-of-the-art in RI uncertainty? Finally, how can any statement of this nature be made when this manuscript does not report measurement uncertainties?

We thank the reviewer to point this out.

Yes, we agree with the reviewer that without providing the refractive index data and measurement uncertainties, we cannot state that "much lower uncertainty than has been achieved previously". We used the AAC instead of the DMA to select particles prior to optical measurement, which avoids the multiple charge issue and thus reducing the uncertainty in the RI derivation.

Thus, in the revised introduction, the statement "thus enabling optical properties – in particular refractive index – to be determined with much lower uncertainty than has been achieved previously" is removed.

In addition, we modified this statement to: "Careful consideration of the impacts of multiply charged particles on subsequent RI derivations can go some way to reducing uncertainty in the resultant RI, but this nevertheless remains a significant contributor to uncertainty (Cotterell et al., 2020;Zarzana et al., 2014;Miles et al., 2011). Thus, the classification of particles without relying on electrical charge should reduce the uncertainty in refractive index retrievals from measured aerosol optical properties. Important additional considerations in the retrieval of refractive indices from optical spectroscopy data are the aerosol morphology (described above) and mixing state. The mixing state can be probed using the Single Particle Soot Photometer (SP2), which can measure the refractory BC (rBC) mass content and optical

size of individual particles. However the SP2 needs an empirical calibration to retrieve the rBC mass from the incandescence signal (Laborde et al., 2012a). The conventional method to calibrate the incandescence channel of SP2 is using size selected Aquadag standards (Acheson Inc. USA) and then correcting to a calibration representative of ambient rBC by a constant factor of 0.75 (Laborde et al., 2012b). However, few experiments since have independently verified this across various soot types."

**Line 118-132 on page 5 and 6.**

The SP2 has been thoroughly described in the literature. Is all of section 2.1.3 necessary? At this stage in the maturation of the SP2, I believe it is more constructive to discuss the artifacts and other measurement caveats of the SP2, for example, Sedlacek et al. (2018). How were these charring artifacts handled in data analysis analysis?

**We thank the reviewer to point this out.**

The sentence of "The SP2 consists of an intense intra-cavity laser operating at a wavelength of 1064 nm and four optical detectors. Particles were drawn into the measurement chamber through a capillary, restricted with a particle free sheath flow and focused into a jet. The particle jet passed through the center of an intra-cavity laser beam operating at 1064 nm." is removed in the revised manuscript.

The sentence is modified to: "The refractory black carbon (rBC) mass concentration was measured by a SP2 (Droplet Measurement Technologies, Colorado, USA). The SP2 uses the laser-induced incandescence to measure the rBC mass and optical size of individual BC particles with an intra cavity Nd:YAG laser operating at 1064 nm."

**Line 226-229 on page 10.**

More discussion regarding the artifacts of the SP2 is added in the revised manuscript.

"For those particles which contain absorbing materials such as the refractory BC, they will absorb 1064 nm light and then heat up and emit visible thermal radiation (incandesce). This incandescence signal is directly proportional to the mass of rBC as determined by a calibration (Liu et al., 2010) with generated BC aerosols of known or independentlymeasured mass. In the atmosphere, besides BC, other materials (e.g., some metals and minerals) can incandesce at 1064 nm as well. However, as boiling point temperatures of these materials are rather different to that of black carbon, it is easy to distinguish them in measurements made using the SP2 equipped with an additional narrowband incandescence detector, such as the one used here (Liu et al., 2018). Recently, Sedlacek et al. (2018) demonstrated that charring of light-absorbing organic particles at 1064 nm can produce the refractory black carbon and then overestimate the rBC mass concentration, however this does not affect pure BC particles and we saw no evidence for an incandescence signal associated with the pure organic particles measured in this study. We investigated the effectiveness of this calibration from different types of BC particles in this study and report the outcomes of these investigations in Sect. 3.3."

**Line 238-252 on Page 10 and 11.**

The SMPS has an even longer history of prior publication. Section 2.1.6 can be significantly shortened to include only the operational parameters such as scan time and flow rates, and the resulting bounds of the size distribution you are able to measure. Once this has been described, the manuscript would benefit from a discussion on the uncertainties introduced by the SMPS. Finally, please describe the inversion algorithm used. Is it the typical inversion algorithm that comes with the SMPS software, or is it custom? If it is custom, it should be discussed and cited.

The section 2.1.6 is modified to simply state: "Aerosol size distributions in the diameter range from 14.9 to 673.2 nm were measured by a commercial SMPS (TSI, USA, employing a model 3082 classifier unit, 3081 DMA and 3786 CPC) with sheath and sample flow rates of 3 L min-1 and 0.3 L min-1, respectively. The DMA was operated in scanning mode with a scan time of 60 s and a retrace time of 4 s. The particle size distribution was corrected for multiple charge effects and diffusion loss using the standard inversion algorithm in the SMPS software (AIM version 10). Before the experiment, the SMPS was calibrated using NIST certified polystyrene latex spheres (PSLs, Thermo Fisher Inc.)."

**Line 276-283 on Page 12.**

Why was 180 °C chosen for the thermal denuder? The optimal internal temperature for a thermal denuder is the subject of ongoing research, yet the consensus is that it must be chosen based on some optimization of experimental parameters. In this case, I would guess that this optimization corresponds to complete volatilization of coating materials so that BC alone can

be studied. However, it is never explicitly stated what the reasons are for choosing 180 °C, nor is the method by which the authors arrived at this temperature. For example, Sumlin et al. (2018) describes an approach using four different TD temperatures in an attempt to volatilize different mass fractions of organic matter. Please discuss the approach to choosing 180 °C.

For the purpose of this study, we wanted to remove as much of the organic coating material from the combustion-generated particles as possible and 180 °C is the maximum temperature for our self-built thermal denuder.

The following sentence is added for further clarification.

"In this study, the purpose of the TD is to remove as much of the organic coatings from the combustion-generated particles as possible, rather than probing the volatility properties of the coating. Therefore, all heating zones of the TD were set to their maximum temperature of 180 °C. This upper temperature is lower than that achieved by other commercial TD units and minimises the risks of charring."

Line 304-309 on page 13.

Furthermore on the subject of the TD, the authors state that the inner diameter of the heating zone is 0.15 meters. This is rather large, and it is almost certain that there is a temperature gradient across the radius. The temperature along the central axis may be several degrees colder than at the walls, and aerosol travelling at different points along the radius will experience varying degrees of thermal processing. Was this temperature gradient measured? How does this gradient affect the analysis?

We agree that there is a temperature gradient across the radius of the heating zone in the thermal denuder. The temperature reported here is that at the axial centre of the thermal denuder's heating zone. We calibrated the TD by measuring the temperature at the point of the central line before the experiment. We did not measure the temperature gradient across the radius in this study.

As the purpose of using TD in this study is remove coatings of coated BC particles rather than investigate the volatility of the particles, the temperature gradient would not influence on our results here. The following sentence is added for further clarification:

"The TD was given 30 mins for its temperature to stabilize before sampling. The temperature of the TD was calibrated by measuring the temperature at the axial centre of the denuder's heating zone. In this study, the purpose of the TD is to remove as much of the organic coatings from the combustion-generated particles as possible, rather than probing the volatility properties of the coating. Therefore, all heating zones of the TD were set to their maximum temperature of 180 °C. This upper temperature is lower than that achieved by other commercial TD units and minimises the risks of charring."

**Line 302-309 on page 13.**

Finally, on the subject of the TD, the authors state that the residence time in the heating section was approximately 31 seconds. Given the measurements provided for both the inlet and outlet (0.037 m ID tube) and the heating section (0.15 m ID) it seems that the authors did not account for fluid velocity change within the heating section. If one considers only 1 liter min-1 flow through the 0.037 m ID tube, across the 0.51 m heating section (assuming steady and conservative flow), one arrives at 32.9 s, which is "approximately" 31 s as stated in the manuscript. However, that assumes that fluid velocity is constant throughout the TD – it is not, since the ID in the heating section is different than the inlet and outlet. The mean residence time in the heating section is likely somewhere on the order of nine minutes. Given this, and my previous comment, I am unsure that conclusions from the TD experiment are represented accurately. It is likely that this data can be re-processed, or at least re-interpreted, but the data may have been handled according to an incomplete understanding of the experimental setup. This must be corrected or addressed in the manuscript.

We thank the reviewer to point this out. However, to be absolutely clear, the exact function of the thermal denuder was simply to remove as much coating as possible such that changes in morphology changes after coating and/or humidification could be detected. Given a clear distinction in results could be seen between different experimental runs, the set-up used was evidently successful in this regard, and it is not clear to us how the data could be reprocessed or reinterpreted in light of a more accurate estimate of the residence time.

That said, we have reviewed the setup of the TD, and added the following sentences for further clarification:

"The temperature in the heating section  $(0.51 \text{ m} \times 0.15 \text{ m} \text{ ID})$  was controlled by four PID controllers (Watlow EZ-ZONE) with additional temperature sensors on the outside of the tube. It is necessary to ensure flow through the TD whenever it is heated, even when bypassed to allow measurements of the unheated sample. A constant 2-2.5 L min-1 flow of the sample air through it. A vacuum line maintained 2.0-2.5 L min-1 through whichever of the bypass or TD line was not in use. The residence time of the air sample in the heating section was 216-270 s. The TD was given 30 mins for its temperature to stabilize before sampling. The temperature of the TD was calibrated by measuring the temperature at the axial centre of the denuder's heating zone. In this study, the purpose of the TD is to remove as much of the organic coatings from the combustion-generated particles as possible, rather than probing the volatility properties of the coating. Therefore, all heating zones of the TD were set to their maximum temperature of 180 °C. This upper temperature is lower than that achieved by other commercial TD units and minimises the risks of charring."

Line 296-309 on page 13.

It seems that all measurements are reported without uncertainties. The authors should report uncertainties or justify their exclusion.

The uncertainties were analysed and added in the Table 1, Figure 4 and Figure 7 in the revised manuscript.

Finally, please make sure all references are properly formatted and include DOI numbers, if possible.

The references are reformatted with DOI numbers included.

**Minor questions and comments**

The name "Manchester Aerosol Chamber" is unfortunate, since it shares an acronym (MAC) with a commonly studied aerosol optical parameter, the mass absorption cross-section. Unless this acronym has been used for your chamber multiple times in other publications, I would suggest you consider renaming it.

We have deleted the acronym (MAC) in the manuscript as we only used it once. The acronym (MAC) in the Fig. 1 is replaced by the "Manchester Aerosol Chamber".

Several parts of the Experimental Methods section contain results, especially section 2.3.4. I suggest re-writing these sections to focus on the methods, and thoroughly discuss the results in section 3.

We have re-arranged the section 2.3 and moved the results in the section 3.

The headings for sections 2.1.1, 2.1.2, 2.1.3, 2.1.5 and 2.1.6 should be spelled out, for consistency with other 2.1.X section headings.

**This is revised.**

The description about the AAC in section 2.2.1 is overly subjective. Furthermore, the AAC only produces a monodisperse aerosol in the sense of aerodynamic diameter. This is not "truly" monodisperse, since there are other measures of aerosol diameter that run contrary to this qualifier.

The term "truly" is removed and the sentence for AAC description is revised to: "The AAC (Cambustion Ltd, Cambridge, UK) is used to select aerosols within a narrow range of aerodynamic diameters and does not suffer from the issue of multiple charges that affects selection using instruments such as the CPMA and DMA."

Line 180-182 on page 8.

Line 278: The authors may consider stating the material that the space blanket (mylar, I'm guessing) is made from so that readers unfamiliar with the term will know what was used.

Yes, the material is mylar. This information is added in the revised manuscript.

Line 281: How was the illumination evaluated?

The illumination on the chamber was evaluated using both spectral radiometry and steadystate actinometry experiments. Briefly, the ambient environment daytime solar spectrum was first measured by spectral radiometry. The photolysis rate of NO2 ( $jNO_2$ ) was estimated in steady-state actinometry used as a confirmation of the light intensity in the chamber (Hu et al., 2014) measured by direct spectral radiometry. Such actinometric measurements were carried out by injecting ~ 70 ppb NO2 into the chamber and irradiating for more than 3 hours, measuring the concentration of NO, NO2 and O3 continuously. The temperature and relative humidity was maintained at around 25°C and 50% respectively. More details can be found in section 3.3 in Shao et al. (2021). Fig. 4 in Shao et al. (2021) shows the total actinic flux measured in the Manchester Atmospheric Chamber (red line) multiplied by 3.5 compared with the Manchester midday clear sky measurements on a June day.

The reference of Shao et al. (2021) is cited in the revised manuscript.

Figure 4: Total actinic flux spectrum in the MAC compared to the ambient light spectrum obtained in the city of Manchester (UK) mid-day with a clear sky in June 2015.

**Reference:**

Shao, Y., Wang, Y., Du, M., Voliotis, A., Alfarra, M. R., Turner, S. F., et al., 2021. Characterisation of the Manchester Aerosol Chamber facility. Atmos. Meas. Tech. Discuss. 2021, 1-50.

Line 357: Can the authors comment on what effect resin content might have on the resulting aerosols?

Many factors can influence the resulting aerosols, such as resin content, moisture content. In this study, we only focus on the influence of water content of wood samples on the physical and chemical properties of the particles from the wood combustion.

The following sentence is added in line 411-419 on page 18.

"Both Scots pine and Western red cedar are resin rich, while the Giant Redwood contains less resin, and the Poplar contains barely any resinous compounds. Hence, all should be capable of producing different volumes and types of aerosol particles. There are many factors that can influence composition and properties of generated aerosols during combustion, such as the wood resin content and moisture content, which would result in a highly extensive variable set. In the work presented here, we only focus on the influence of water content of wood samples on the physical and chemical properties of the particles from wood combustion. We assessed wood of two different fuel moistures: fully oven dried samples and moist samples with  $\sim 25\%$  moisture content. "

Line 366-367: How was the "sufficient concentration" to ignite the pyrolysate determined?

During the experiment, after the wood samples were placed under the radiant heat flux, a spark igniter was positioned in the released stream of the pyrolysate to attempt to ignite the wood samples. Immediately following the initial placement of the wood sample under the radiant heat source, pyrolysate is continuously released. "Sufficient concentration" refers to the pyrolysate concentration that is high enough to enable ignition of the wood sample. We have no metric to quantify this pyrolysate concentration.

The following sentence is added in line 423-430 on page 18 and 19 for further clarification:

"A continuously operated spark igniter that was positioned in the released stream of pyrolysate acted as a source of ignition. Immediately following the initial placement of the wood sample under the radiant heat source, pyrolysate was continuously released and the pyrolysate concentration immediately above the wood sample and in the vicinity of the spark igniter increased also. The sample ignited once this pyrolysate reached sufficient concentrations and was well mixed with the surrounding air, at which point the igniter was switched off and removed from the air flow."

Line 380: Change "exemplar" to "example".

**This is revised.**

Line 404: Change "300 s of the ignition" to "300 s after the ignition."

This is revised.

Line 430: I suggest specifying the  $NO_2$  came from a "compressed gas cylinder". Also, what was the concentration of  $NO_2$ , and what was the balance gas?

The sentence is modified to: "During the experiment,  $NO_2$  (10% v/v, with a balance gas of high purity  $N_2$  (BOC, UK)) was injected directly into the bag from a custom-made gas cylinder via stainless steel tubing, and its concentration was measured using a chemi-luminescence gas analyzer (Model 42i, Thermo Scientific, MA, USA)."

Line 459-462 on page 20.

Line 432: Change "desirable" to "desired".

**This is revised.**

Lines 451-454: I suspect that the increase in ozone concentration is attributed to the absorption of UV photons by  $O_2$  in the chamber – this is a common method to produce  $O_3$  from  $O_2$ . I cannot find any references in the literature on O3 production due to UV absorption by cresol. If the authors can provide a reference for  $O_3$  production by cresol photochemistry, please do so.

As shown in Fig. 3, the increase of ozone concentration occurs only after the injection of the VOC (cresol). At the instance of cresol injection, the lights are off. Therefore, the abrupt increase in ozone concentration detected by the ozone monitor around the time of VOC injection is not caused by photochemistry. The signal detected by the  $O_3$  monitor may be due to cresol absorbing light at the wavelength it uses.

Line 464: Change "desirable" to "desired". What was the desired concentration?

**This is revised.**

Line 470: "particulate" is an adjective. Change to "particles".

**This is revised.**

Line 471-472: What size did the particles stabilize to?

This information is added in the revised manuscript.

"After the condensed organics equilibrate with the surrounding VOCs and the particles stabilised at a certain size (295 nm in aerodynamic size), the aerodynamic particle size distribution of the dried organic coated BC particles, and that of the dried organic coated BC particles experienced the humidity cycling, was measured by AAC. Hereafter, a thermal denuder operated at 180 °C was added, and the organic coatings of the coated BC particles, that had either experienced the humidity cycling process or not, were removed by the TD, and then the size distribution of the BC core was measured by the AAC and SMPS."

**Line 488-495 on page 21.**

The caption on Figure 1 refers readers to the main text to thoroughly understand what is meant by the different groupings of instruments, however, the first reference to the figure in section 2 does not clarify the groupings whatsoever. The figure is therefore unclear: were different groupings of instruments used during different experiments? Were they grouped in this way during data analysis to calculate specific parameters? This needs to be clarified both in the figure and its caption, and in the main text.

In this study, we use the different groupings of instruments to achieve different target. This information is added in the main text, the caption in the Figure 1.

The following sentences are added in the line 170-177 on page 8.

"The right panel in Fig. 1 shows the various instrument configurations used in this study to target different measurements: Instrument set (1) was used for measurements of aerosol optical properties (and thereby enable the retrieval of refractive index for BC/BrC aerosols, subject of a future publication) and organic chemical measurement for wood combustion (setup (a)) and chamber (setup (c)) experiments. Instrument set (2) was used for dynamic shape factor measurements for BC and material density measurement for organic aerosols (setup (a) and (b)). Instrument set (3) was used for SP2 incandescence signal calibration (setup (b)). Instruments set (4) was used for BC restructuring experiments (setup (c))."

The caption in Figure 1 is modified to:

**"Fig. 1.** Schematic diagram of the experimental configuration for: (a) Wood combustion; (b) SP2 incandesces signal calibration and BC morphology investigation; (c) Brown carbon formation and restructuring of BC. The combinations of instruments described in (1), (2), (3)

and (4) represent different measurement configurations used to enable characterisation of specific aerosol physiochemical parameters, as described in the main text. The instrument set (1) was used for measurements of aerosol optical properties (and thereby enable the retrieval of refractive index for BC/BrC aerosols, subject of a future publication) and organic chemical measurement for wood combustion (setup (a)) and chamber (setup (c)) experiments; instrument set (2) was used for dynamic shape factor measurements for BC and material density measurement for organic aerosols (setup (a) and (b)); the instrument set (3) was used for SP2 incandescence signal calibration (setup (b)); the instruments set (4) was used for BC restructuring experiments (setup (c))."

**References**

Sedlacek et al., Formation of refractory black carbon by SP2-induced charring of organic aerosol, Aerosol Sci. & Tech, 52:12, 1345-1350, 10.1080/02786826.2018.1531107, 2018.

Sumlin et al., Density and Homogeneous Internal Composition of Primary Brown Carbon Aerosol, Environ. Sci. Technol. 2018, 52, 7, 3982-3989, 10.1021/acs.est.8b00093, 2018.